# LRP1 integrates murine macrophage cholesterol homeostasis and inflammatory responses in atherosclerosis

Xunde Xian[1][†]*, Yinyuan Ding[1,2][†], Marco Dieckmann[1][†], Li Zhou[1], Florian Plattner[3,4], Mingxia Liu[5], John S Parks[5], Robert E Hammer[6], Philippe Boucher[7], Shirling Tsai[8,9], Joachim Herz[1,4,10,11]*

[1]Departments of Molecular Genetics, UT Southwestern Medical Center, Dallas, United States; [2]Key Laboratory of Medical Electrophysiology, Ministry of Education of China, Institute of Cardiovascular Research, Southwest Medical University, Luzhou, China; [3]Department of Psychiatry, University of Texas Southwestern Medical Center, Dallas, United States; [4]Center for Translational Neurodegeneration Research, University of Texas Southwestern Medical Center, Dallas, United States; [5]Section on Molecular Medicine, Department of Internal Medicine, Wake Forest School of Medicine, Winston-Salem, North Carolina; [6]Department of Biochemistry, University of Texas Southwestern Medical Center, Dallas, United States; [7]CNRS, UMR 7213, University of Strasbourg, Illkirch, France; [8]Department of Surgery, UT Southwestern Medical Center, Dallas, United States; [9]Dallas VA Medical Center, Dallas, United States; [10]Department of Neuroscience, UT Southwestern, Dallas, United States; [11]Department of Neurology and Neurotherapeutics, UT Southwestern, Dallas, United States

*For correspondence:
xunde.xian@utsouthwestern.edu
(XX);
joachim.herz@utsouthwestern.edu
(JH)

[†]These authors contributed
equally to this work

**Competing interests:** The
authors declare that no
competing interests exist.

**Reviewing editor:** Gary L
Westbrook, Vollum Institute,
United States

**Abstract** Low-density lipoprotein receptor-related protein 1 (LRP1) is a multifunctional cell surface receptor with diverse physiological roles, ranging from cellular uptake of lipoproteins and other cargo by endocytosis to sensor of the extracellular environment and integrator of a wide range of signaling mechanisms. As a chylomicron remnant receptor, LRP1 controls systemic lipid metabolism in concert with the LDL receptor in the liver, whereas in smooth muscle cells (SMC) LRP1 functions as a co-receptor for TGFβ and PDGFRβ in reverse cholesterol transport and the maintenance of vascular wall integrity. Here we used a knockin mouse model to uncover a novel atheroprotective role for LRP1 in macrophages where tyrosine phosphorylation of an NPxY motif in its intracellular domain initiates a signaling cascade along an LRP1/SHC1/PI3K/AKT/PPARγ/LXR axis to regulate and integrate cellular cholesterol homeostasis through the expression of the major cholesterol exporter ABCA1 with apoptotic cell removal and inflammatory responses.
DOI: https://doi.org/10.7554/eLife.29292.001

## Introduction

Atherosclerosis is a chronic condition characterized by impairments in three major processes: systemic and cellular cholesterol homeostasis, inflammation, and apoptosis/efferocytosis (*Libby et al., 2011*). Essential roles for the multifunctional transmembrane protein LDL receptor-related-protein 1 (LRP1) have been reported for each of these three processes (*Boucher et al., 2003*; *Boucher et al., 2002*; *El Asmar et al., 2016*; *Mantuano et al., 2016*; *Subramanian et al., 2014*; *Zhou et al., 2009a*; *Zurhove et al., 2008*). Together with LDLR, LRP1 regulates the clearance of circulating cholesterol-rich remnant proteins by hepatocytes (*Rohlmann et al., 1998*). LRP1 is also a key regulator

**eLife digest** Atherosclerosis is a disease in which "plaques" build up inside the walls of arteries. Plaques consist of a fatty substance called cholesterol, together with immune cells such as macrophages and other material from the blood. Over time, the plaque narrows and hardens the arteries. This restricts the flow of blood to vital parts of the body, which increases the risk of heart attacks, strokes and other severe conditions.

Macrophages play an important role in atherosclerosis. At the early stage of the disease, macrophages enter the developing plaques to take up the excess cholesterol. Cholesterol taken up by macrophages needs to be exported out of the cell and sent to the liver for removal. Yet, these processes can go awry. Macrophages can fill up with too much cholesterol and become trapped in the arteries. These cholesterol-laden macrophages can also start dying. These problems enable the plaques to grow and worsen the disease.

LRP1 is an important protein present on the surface of many types of cells. In macrophages, LRP1 helps to export excess cholesterol out of the cell, thus lowering the risk of atherosclerosis. LRP1 also reduces cell death in the plaque, which slows the plaques' progression. Previous research has shown that the region of LRP1 present inside the cell can be modified by the attachment of a phosphate group – a process termed phosphorylation. Whether phosphorylation of LRP1 plays a role in preventing atherosclerosis is not understood.

To address this question, Xian, Ding, Dieckmann et al. engineered mice in which LRP1 was unable to get phosphorylated. The results show that phosphorylated LRP1 – but not the non-phosphorylated version – turns on a signaling pathway in macrophages. This pathway increases the expression of a transporter protein that exports cholesterol out of the cell. This reduces the amount of cholesterol that accumulates in macrophages. Lastly, mice with problems with LRP1 phosphorylation developed more severe atherosclerotic plaques with more dying cells present in the affected areas compared to normal mice.

These findings show how phosphorylation of LRP1 protects against atherosclerosis. Understanding this process in further detail may help scientists to devise new ways to treat this disease.

DOI: https://doi.org/10.7554/eLife.29292.002

of intracellular cholesterol accumulation in macrophages and smooth muscle cells (SMCs) (*Boucher et al., 2003*; *Lillis et al., 2015*; *Zhou et al., 2009a*). In SMCs, LRP1 functions as a co-receptor with platelet-derived growth factor receptor (PDGFRβ), which in turn activates inflammatory and pro-thrombotic processes (*Boucher et al., 2002*; *Loukinova et al., 2002*). LRP1 itself has also been implicated in limiting cellular inflammatory responses (*Mantuano et al., 2016*; *Zurhove et al., 2008*), and the dual role of LRP1 in cholesterol homeostasis and inflammation (*El Asmar et al., 2016*; *Woldt et al., 2011*, *2012*) was recently demonstrated in an adipocyte-specific LRP1-deficient mouse model (*Konaniah et al., 2017*). Finally, LRP1 is necessary for efficient efferocytosis in macrophages and dendritic cells (*Subramanian et al., 2014*; *Yancey et al., 2010*).

LRP1 is ubiquitously expressed, but how it affects cellular signaling mechanisms can differ substantially in a cell-type dependent manner. In SMCs, LRP1 regulates ERK1/2 activation, which leads to increased cPLA2 phosphorylation and release of arachidonic acid (*Graves et al., 1996*; *Lin et al., 1992*), a suppressor of LXR-driven ATP-binding cassette transporter A1 (ABCA1) expression (*DeBose-Boyd et al., 2001*), thereby increasing SMC intracellular cholesterol accumulation (*Zhou et al., 2009a*). ABCA1, however, mediates not just reverse cholesterol export, it also regulates cellular inflammatory responses (*Ito et al., 2015*; *Tang et al., 2009*; *Zhu et al., 2010*). This suggests the presence of an LRP1/LXR/ABCA1 axis that controls and integrates cellular cholesterol export and inflammatory responses.

One of the atheroprotective roles of LRP1 in the vascular wall is the regulation of the well-established mitogenic pathway mediated by platelet derived growth factor (PDGF), which promotes atherosclerosis (*Ross, 1993*). Mitogenic signaling is commonly associated with tyrosine phosphorylation, and stimulation of PDGFRβ with PDGF-BB induces tyrosine phosphorylation of LRP1 by Src-family tyrosine kinases at the distal cytoplasmic NPxY motif (*Barnes et al., 2001*;

*Boucher et al., 2002*; *Loukinova et al., 2002*). This phosphorylation event can either promote or inhibit the association of the LRP1-ICD (intracellular domain) with several intracellular adaptor proteins, which in turn modulates down-stream signaling cascades (*Gotthardt et al., 2000*). However, the physiological significance of LRP1 NPxY phosphorylation has remained unclear.

To investigate this role of LRP1 tyrosine phosphorylation in atherosclerosis, we developed a knock-in mouse model in which the tyrosine in the distal NPxY motif was replaced with phenylalanine (*Lrp1$^{Y63F}$*), thereby disabling Lrp1 NPxY phosphorylation while leaving the interaction with phospho-tyrosine-independent PTB-domain containing adaptor proteins intact (*Gotthardt et al., 2000*; *Trommsdorff et al., 1998*). We found that LRP1 NPxY phosphorylation induces a molecular switch of associated cytoplasmic adaptor proteins away from endocytosis-promoting DAB2 to the well-known signal-transducer SHC1. SHC1 is an evolutionarily conserved adaptor protein that mediates mitogenic signaling (*Dieckmann et al., 2010*; *Pelicci et al., 1992*; *van der Geer, 2002*; *van der Geer and Pawson, 1995*; *van der Geer et al., 1995*) and is required for LRP1-dependent signal transduction in SMC through activation of PI3K and phosphorylation of AKT (*Gu et al., 2000*; *Radhakrishnan et al., 2008*). Surprisingly, lack of LRP1 tyrosine phosphorylation was inconsequential in SMCs. By contrast, in macrophages, SHC1 binding to phosphorylated LRP1 activated PI3K/AKT and PPARγ/LXR driven ABCA1 expression, a major cellular cholesterol exporter, and one of many mechanisms that is impaired in atherogenesis. Consequently, disabling LRP1 tyrosine phosphorylation resulted in enhanced macrophage intracellular lipid accumulation and decreased clearance of apoptotic cells, thus resulting in accelerated atherosclerosis independent of plasma cholesterol levels.

## Results

### Generation and characterization of *Lrp1$^{Y63F}$* mice

To investigate the role of LRP1 tyrosine phosphorylation in atherogenesis, we generated a knock-in mouse that was incapable of tyrosine phosphorylation at the distal cytosolic NPxY motif at cytoplasmic amino acid position 63 (*Lrp1$^{Y63F}$*) and then backcrossed it into an Ldlr-deficient background (*Lrp1$^{Y63F}$;Ldlr$^{-/-}$*; *Figure 1A*). Western blot analysis of liver and aortic extracts from *Lrp1$^{Y63F}$* mice revealed no effect of the Y63F mutation on Lrp1 expression in the liver or the aorta (*Figure 1B*). Pull-down experiments in SMC stimulated with PDGF-BB confirmed decreased phosphorylation of the Lrp1 cytoplasmic domain in *Lrp1$^{Y63F}$* SMC when compared with wild type SMC (*Figure 1C*). The LRP1 cytoplasmic domain interacts with numerous intracellular adaptor and scaffold proteins (*Gotthardt et al., 2000*; *Herz and Strickland, 2001*; *Stockinger et al., 2000*). Using co-immunoprecipitation, we observed differential binding of intracellular adaptor proteins Dab2 (Disabled-2) and Shc1 (SHC-transforming protein 1) to Lrp1 depending on the phosphorylation state of the NPxY motif. After stimulation of wild type SMC with PDGF-BB, which induces phosphorylation of the distal NPxY motif in the Lrp1 cytoplasmic domain, Shc1 was bound to Lrp1, but Dab2 was not. In *Lrp1$^{Y63F}$* SMC, in which phosphorylation of the distal NPxY motif is prevented, Shc1 was no longer able to bind to Lrp1, while Dab2 could now be co-immunoprecipitated with Lrp1 (*Figure 1C*). Thus, binding of Shc1 to Lrp1 is regulated by tyrosine phosphorylation of the distal NPxY motif of Lrp1.

LDLR and LRP1 have functions in systemic lipoprotein metabolism. We therefore analyzed the effect of high cholesterol/high fat (HCHF) diet feeding on *Lrp1$^{Y63F}$;Ldlr$^{-/-}$* mice. Body weight, plasma cholesterol and triglyceride concentrations of *Lrp1$^{Y63F}$;Ldlr$^{-/-}$* mice on chow diet were essentially identical to age and sex-matched *Ldlr$^{-/-}$* controls (*Figure 2A*). However, after 16 weeks on HCHF diet, *Lrp1$^{Y63F}$;Ldlr$^{-/-}$* mice had significantly increased body weight, plasma cholesterol (2035 ± 99.53 mg/dl vs. 1672 ± 45.98 mg/dl, p<0.01) and triglyceride levels (257.4 ± 26.2 mg/dl vs. 161.8 ± 13.55 mg/dl, p<0.01) compared to *Ldlr$^{-/-}$* mice (*Figure 2A*). Western blot analysis revealed increased ApoB48 and ApoE in *Lrp1$^{Y63F}$;Ldlr$^{-/-}$* mice, whereas ApoB100 and ApoAI were decreased compared to *Ldlr$^{-/-}$* mice after HCHF diet feeding (*Figure 2B*). FPLC lipoprotein analysis demonstrated that elevated plasma cholesterol levels in *Lrp1$^{Y63F}$;Ldlr$^{-/-}$* mice fed with HCHF diet were attributed to increased VLDL-cholesterol with slightly reduced LDL- and HDL-cholesterol contents (*Figure 2C*). VLDL-triglyceride levels were also increased in *Lrp1$^{Y63F}$;Ldlr$^{-/-}$* mice (*Figure 2C*). Consistent with the results of total plasma apolipoprotein analysis, size-fractionated ApoB100 and ApoAI were significantly decreased, but ApoE was increased in *Lrp1$^{Y63F}$;Ldlr$^{-/-}$* mice (*Figure 2D*).

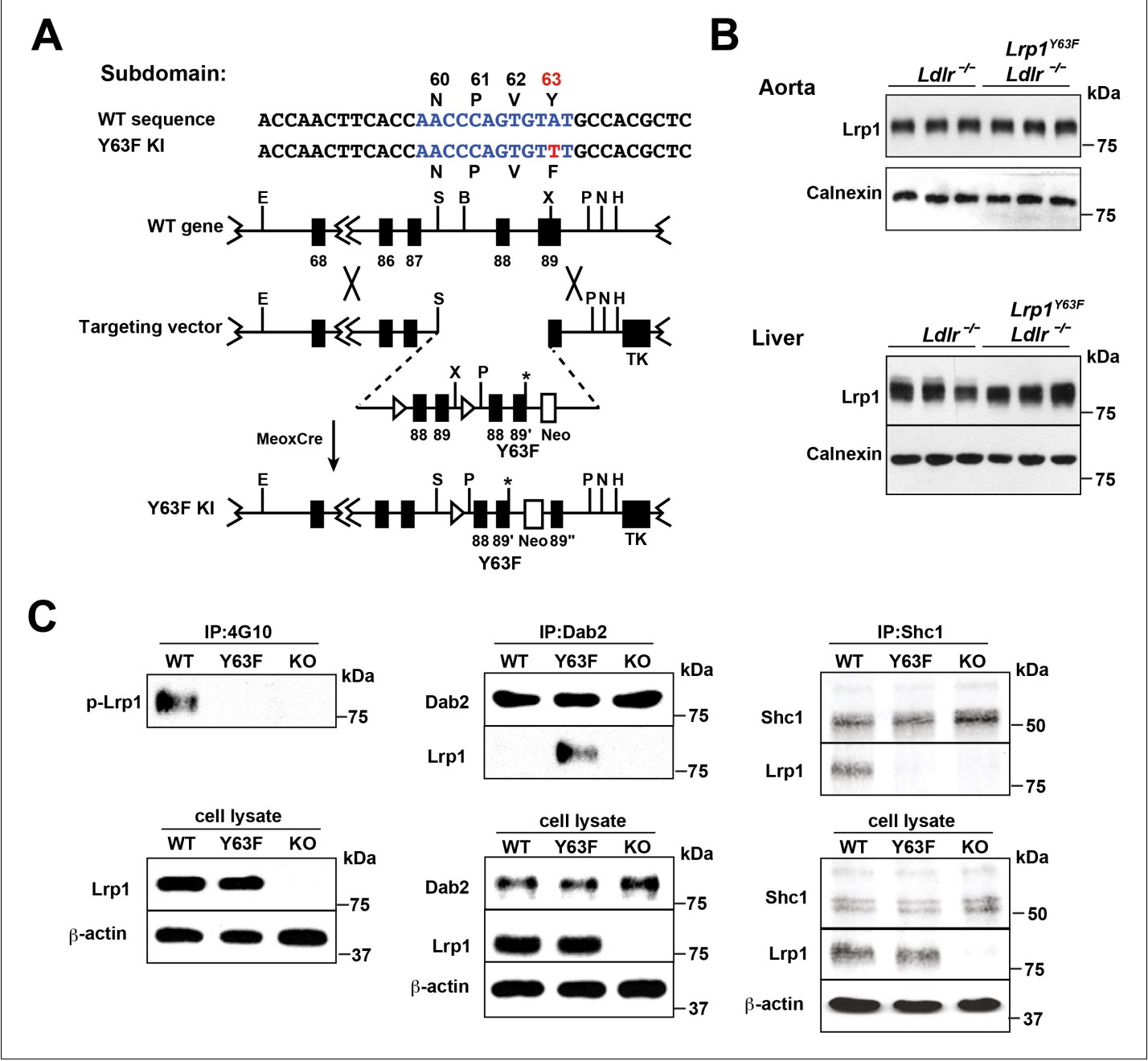

**Figure 1.** Generation and characterization of *Lrp1^Y63F* knock-in mouse model. (**A**) Schematic of the construct used to generate *Lrp1^Y63F* knock-in mice with a point mutation in the tyrosine phosphorylation site (Y63F) as indicated in red and asterisk. Open triangles indicate loxP sites. The neomycin-resistance (Neo) and thymidine kinase (TK) genes were used for selection of the targeted ES cells. Restriction sites: E, EcoRI; S,SpeI; B,BamHI; X,XhoI; P, PmeI; N,NotI; H,HindIII. (**B**) Western blot analysis of Lrp1 expression levels in aorta and liver from *Lrp1^Y63F*;*Ldlr^−/−* mice (n = 5) and *Ldlr^−/−* controls (n = 5) after 16 week of high cholesterol/high fat (HCHF) diet. (**C**) Impaired Lrp1 binding to Shc1 in primary smooth muscle cells (SMCs) from *Lrp1^Y63F* mice. Primary SMCs were explanted from the aorta of wild type, *Lrp1^Y63F*, or *smLrp1^−/−* mice. Cells were treated with 10 ng/ml PDGF-BB for 10 min. Representative immunoblots probed for LRP1 (upper) and total protein levels in input cell lysate (lower) from co-immunoprecipitation assays using antibodies against phospho-tyrosine (4G10) (left), Dab2 (middle) or Shc1 (right). p-Lrp1, tyrosine phosphorylated Lrp1. Calnexin and β-actin were used as loading controls.

DOI: https://doi.org/10.7554/eLife.29292.003

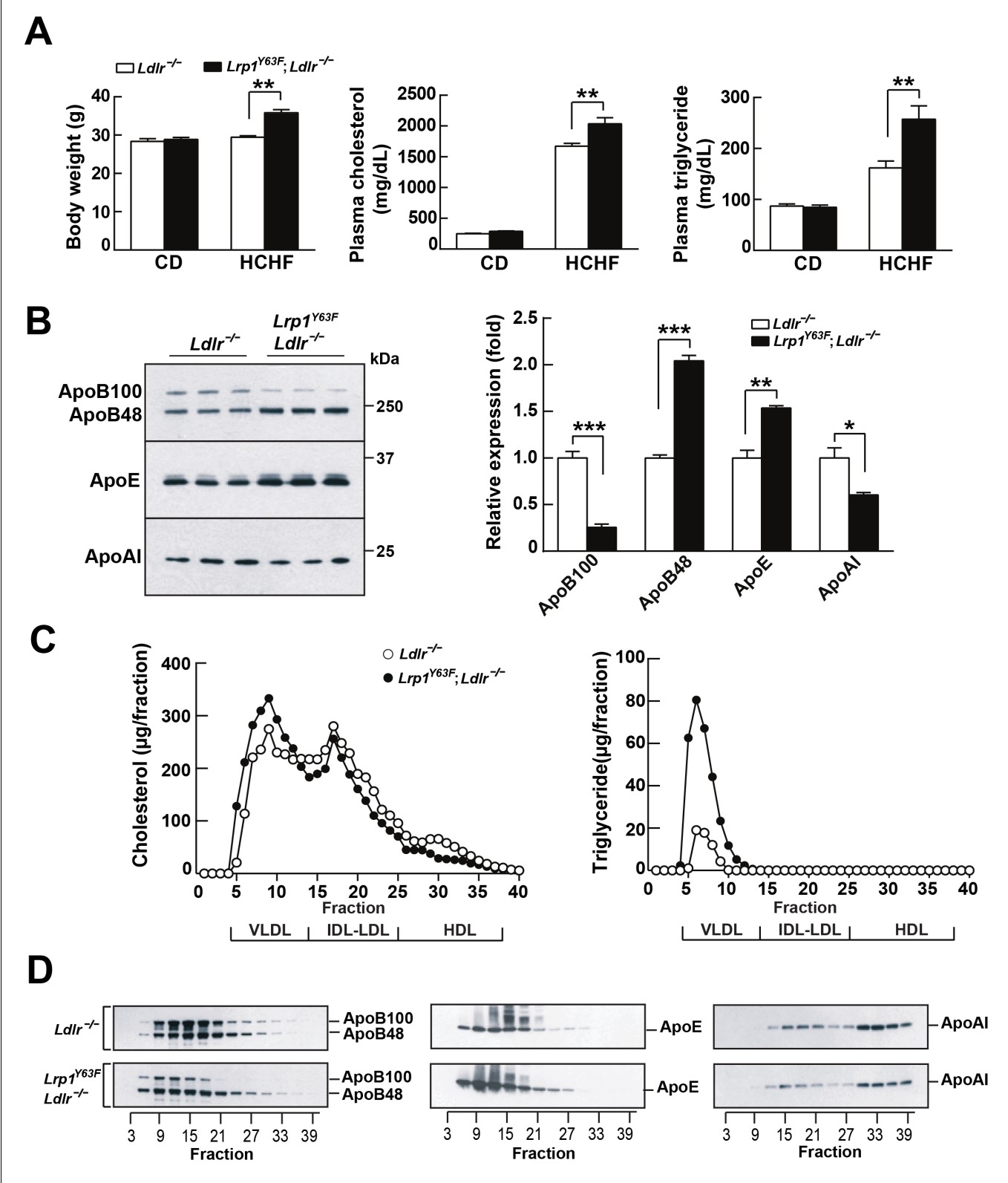

**Figure 2.** *Lrp1^{Y63F}* mutation elicits pro-atherogenic lipoprotein profiles on HCHF diet. (**A**) Body weight, plasma cholesterol and triglyceride levels of *Lrp1^{Y63F}*;*Ldlr^{−/−}* (n = 16) and *Ldlr^{−/−}* (n = 14) mice after 16 weeks of chow diet (CD) or high cholesterol/high fat (HCHF) diet. (**B**) Representative immunoblots and quantitative analysis of plasma apolipoproteins in *Lrp1^{Y63F}*;*Ldlr^{−/−}* (n = 6) and *Ldlr^{−/−}* (n = 6) mice after 16 weeks of HCHF diet. (**C**)

*Figure 2 continued on next page*

*Figure 2 continued*
FPLC plasma lipid profiles. Cholesterol (left) and triglyceride (right) in fractionated plasma from mice on 16 week HCHF diet. (D) Representative immunoblots of ApoB (left), ApoE (middle) and ApoAI (right) in different fractions. All data are mean ±SEM. *p<0.05, **p<0.01, ***p<0.001.
DOI: https://doi.org/10.7554/eLife.29292.004

Interestingly, ApoB48 remnant particles were shifted towards a larger size in $Lrp1^{Y63F};Ldlr^{-/-}$ mice compared to $Ldlr^{-/-}$ mice on HCHF diet, possibly suggesting impaired clearance through SR-B1 (*Rigotti et al., 1997*). We conclude that the $Lrp1^{Y63F}$ mutation not only promotes the accumulation of ApoB48 remnants, as also seen in the hepatic $Lrp1$ knockout (*Rohlmann et al., 1998*; *Rohlmann et al., 1996*), but also causes decreases in HDL and ApoA1, which could independently increase atherosclerosis reminiscent of the increased risk of premature atherosclerosis in individuals with Tangier's disease (*Tall and Wang, 2000*).

## Accelerated atherosclerosis in Lrp1$^{Y63F}$;Ldlr$^{-/-}$ mice

The unexpected, though modest, change in plasma ApoA1 in addition to the accumulation of very large lipoprotein remnants led us to investigate atherosclerosis susceptibility in $Lrp1^{Y63F};Ldlr^{-/-}$ mice. Compared to $Ldlr^{-/-}$ mice, $Lrp1^{Y63F};Ldlr^{-/-}$ mice showed a 1.7-fold and 2.7-fold increase of atherosclerotic lesion area throughout the entire aorta and in the aortic root, respectively (*Figure 3A and B*).

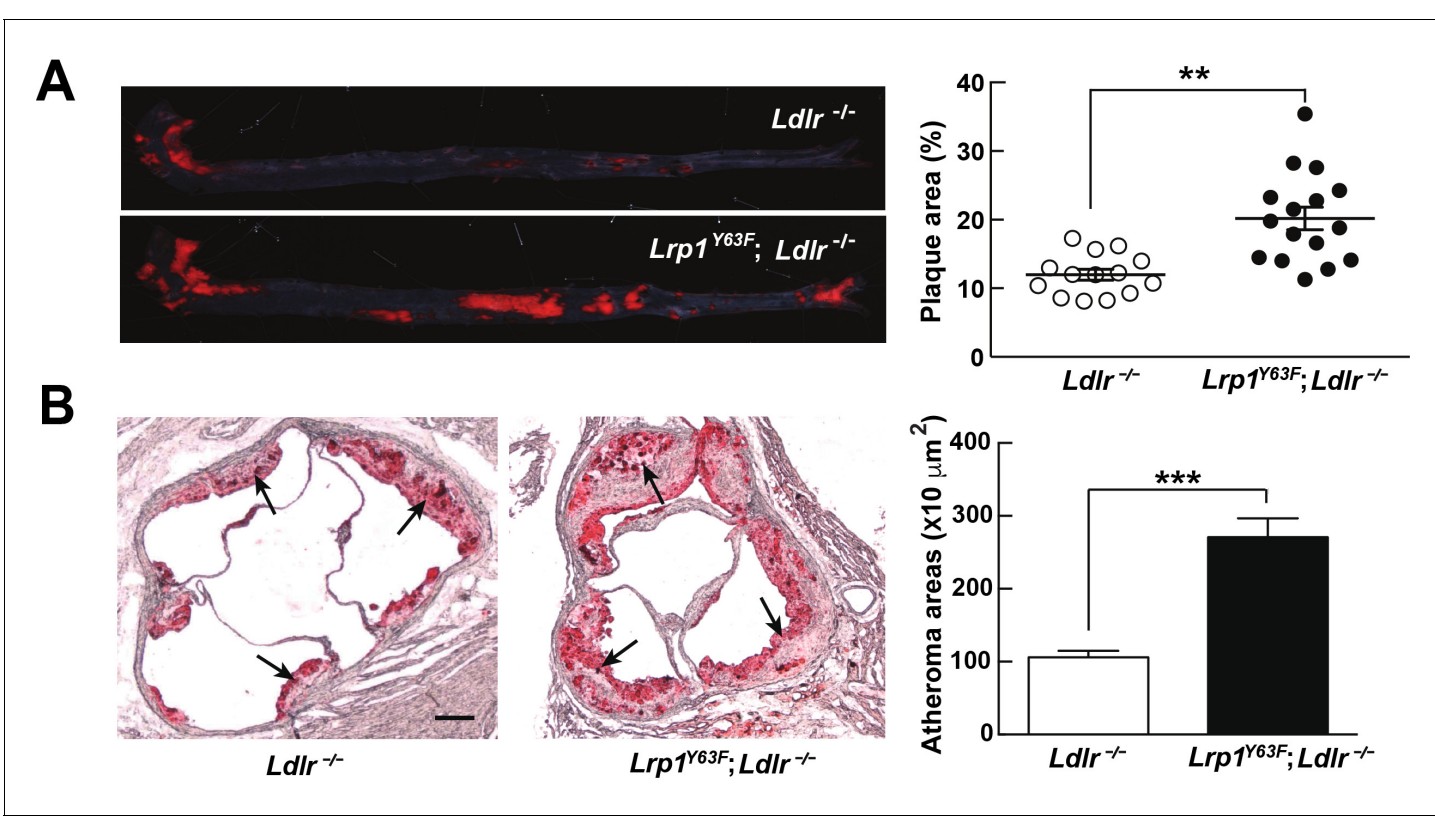

**Figure 3.** $Lrp1^{Y63F}$ mutation accelerates atherosclerotic development on a HCHF diet. (A) Representative images of en face assessment of aortic atherosclerotic lesions stained with Oil red O (left) and quantitative analysis of aortic lesion area (right) in $Lrp1^{Y63F};Ldlr^{-/-}$ (n = 16) and $Ldlr^{-/-}$ (n = 14) mice after 16 weeks of HCHF diet. (B) Analysis of atherosclerotic lesions in aortic roots with Oil red O staining in mice described above. Black arrows indicate atherosclerotic plaque (left, scale bar = 200 μM). Oil red O positive areas in aortic roots were quantified (right). Values are expressed as mean ±SEM. *p<0.05, **p<0.01, ***p<0.001.
DOI: https://doi.org/10.7554/eLife.29292.005

## Lrp1[Y63F] does not alter SMC phenotype or function

Our previous studies demonstrated that smooth muscle Lrp1 deficiency leads to increased SMC proliferation and migration and retention of cholesterol, leading to profound aortic atherosclerosis. Unlike the SMC-targeted *Lrp1* knockout mouse model (*smLrp1*[−/−]), which displays prominent aortic wall thickening and increased numbers of SMC even under normolipidemic conditions, morphometric analysis of the thoracic aorta from *Lrp1*[Y63F];*Ldlr*[−/−] and *Ldlr*[−/−] mice revealed no difference in aortic wall thickness or numbers of SMC (**Figure 4A**). Furthermore, analysis of whole aorta homogenate (**Figure 4B**) and explanted primary aortic SMC (**Figure 4C**) from the *Lrp1*[Y63F];*Ldlr*[−/−] and *Ldlr*[−/−] mice showed no significant differences in the activation of known PDGF down-stream signal effectors. Finally, to test the functional properties of the explanted SMC, we used Transwell Migration and BrdU Proliferation assays to compare their migratory and proliferative response to PDGF-BB. No difference in migration or proliferation was observed between wild type and *Lrp1*[Y63F] SMC (**Figure 4D**). We therefore concluded that the *Lrp1*[Y63F] mutation does not increase atherosclerosis propensity through a change of SMC phenotype.

## Increased necrotic core size and macrophage content in *Lrp1*[Y63F] atherosclerotic lesions

Aside from SMCs, macrophages are central to atherogenesis. Semi-quantitative, cross-sectional analysis of the composition of atherosclerotic lesions in the aortic sinus revealed a 57% increase in Mac-3–positive macrophages, but a comparable amount of α-actin–positive cells (corresponding to SMC) in *Lrp1*[Y63F];*Ldlr*[−/−] mice, consistent with a pronounced infiltration of macrophages into the atherosclerotic plaque (**Figure 5A**). Further semi-quantitative analysis of H&E staining showed a 40% increase in necrotic core area in *Lrp1*[Y63F];*Ldlr*[−/−] mice (**Figure 5B**). The increase in apoptosis was confirmed by TUNEL staining of atherosclerotic plaques in vivo (**Figure 5C**) and oxLDL loaded peritoneal macrophages in vitro (**Figure 5D**). Deficiency in MER proto-oncogene tyrosine kinase (MerTK) is associated with increased macrophage apoptosis and decreased efferocytosis, leading to increased necrotic cores in *Apoe*[−/−] mice (**Li et al., 2006**; **Scott et al., 2001**; **Thorp et al., 2008**). We isolated peritoneal macrophages from *Lrp1*[Y63F];*Ldlr*[−/−] and control mice and found a significant decrease in MerTK expression both at baseline and in response to treatment with oxLDL (**Figure 5E**). Therefore, increased necrotic core size and apoptosis in the *Lrp1*[Y63F];*Ldlr*[−/−] mice may be attributed at least in part to MerTK dysfunction associated with the macrophage *Lrp1* mutation, in addition to other efferocytosis-related functions that are controlled by LRP1 (**Subramanian et al., 2014**; **Yancey et al., 2010**). Based on these results, we suspected that the striking increase in atherosclerosis in *Lrp1*[Y63F];*Ldlr*[−/−] mice is due to macrophages and not SMCs.

## Macrophages drive the atherosclerotic response in Lrp1[Y63F];Ldlr[−/−] mice

Because of the observed change of the lipoprotein profile in the *Lrp1*[Y63F];*Ldlr*[−/−] mice, a trivial explanation for the increased atherosclerosis would be impaired chylomicron clearance and decreased HDL. To exclude the possibility that the atherosclerotic phenotype was secondary to changes in lipid metabolism and to explore the degree to which atherosclerosis was driven by Lrp1 tyrosine phosphorylation in macrophages, we performed bone marrow transplants where *Ldlr*[−/−] or *Lrp1*[Y63F];*Ldlr*[−/−] bone-marrow was transplanted into irradiated *Ldlr*[−/−] mice.

After 16 weeks of HCHF diet all irradiated mouse groups that underwent bone marrow transplant exhibited high plasma cholesterol levels (**Figure 6A**). However, there was no difference in plasma lipid and apolipoprotein levels between *Ldlr*[−/−] → *Ldlr*[−/−] and *Lrp1*[Y63F]; *Ldlr*[−/−] → *Ldlr*[−/−] mice (**Figure 6B, C and D**). By comparing **Figure 6C** to **Figure 2C**, we demonstrated that bone marrow transplantation eliminated the difference in peripheral lipid profiles, thereby demonstrating that the *Lrp1*[Y63F] mutation in macrophages is not responsible for the changes in the lipid profile. Furthermore, the correction of HDL and ApoA1 levels in the bone marrow transplant model suggested a role for Y63 phosphorylation in the liver. Increased catabolism of ApoA1 is a hallmark of the functional impairment of ABCA1 in Tangier's disease (**Tall and Wang, 2000**) and in liver *Abca1* deficient mice (**Timmins et al., 2005**). We therefore investigated the effect of the *Lrp1*[Y63F] mutation in the liver. After a 16 week HCHF diet challenge, Western blot analysis showed decreased expression of Abca1 in the liver from *Lrp1*[Y63F];*Ldlr*[−/−] mice (**Figure 7A**), but no change in Abca1 expression in the

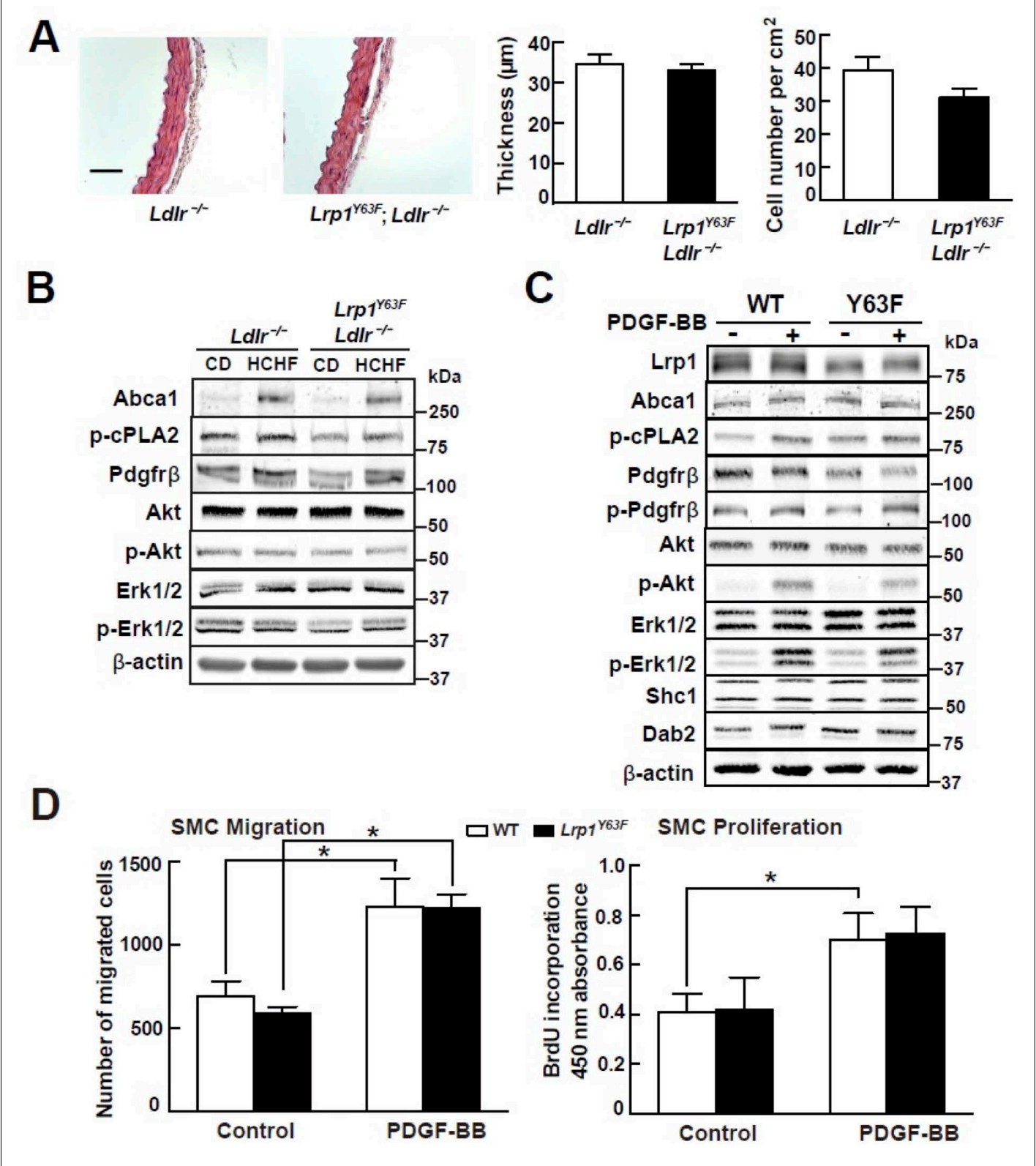

**Figure 4.** *Lrp1*[Y63F] mutation has no effect on atherogenic signaling pathways and morphological phenotype in SMCs. (A) Left: representative images of H&E stained descending thoracic aortas (scale bar = 50 μm). Right: aortic wall thickness and cell number were quantified for the indicated genotypes. Data are expressed mean ±SEM; n = 3 per group and genotype. (B) HCHF diet in *Lrp1*[Y63F];*Ldlr*[−/−] mice has no effect on PDGF and TGFβ-mediated signaling in the aorta. Proteins were extracted from whole aorta in *Lrp1*[Y63F];*Ldlr*[−/−] and *Ldlr*[−/−] mice fed with indicated diets. Expression of proteins

*Figure 4 continued on next page*

Figure 4 continued

involved in PDGF signaling pathways was determined by Western blot analysis. (C) $Lrp1^{Y63F}$ mutation did not affect atherogenic signaling pathways mediated by PDGF in vitro. Primary SMCs from WT and $Lrp1^{Y63F}$ mice were incubated with or without 10 ng/ml PDGF-BB for 10 min. Expression of proteins involved in PDGF signaling were determined by Western blot analysis. (D) $Lrp1^{Y63F}$ mutation did not alter SMC functions in vitro. Primary cultured SMCs from WT and $Lrp1^{Y63F}$ mice were incubated with or without 10 ng/ml PDGF-BB under indicated conditions, then cell migration (left) and proliferation (right) were determined. Values from three independent experiments are expressed as mean ±SEM. *p<0.05.
DOI: https://doi.org/10.7554/eLife.29292.006

liver of $Lrp1^{Y63F};Ldlr^{-/-} \rightarrow Ldlr^{-/-}$ mice (Figure 7B). Therefore, as in Tangier's disease, the significant decrease in hepatic Abca1 likely accounts for the decreased plasma ApoA1 in the $Lrp1^{Y63F}$ mouse. However, even though hepatic Abca1 expression (and therefore plasma ApoA1 levels) is normalized in the bone marrow transplantation model, $Lrp1^{Y63F};Ldlr^{-/-} \rightarrow Ldlr^{-/-}$ still developed more atherosclerotic lesions when compared to $Ldlr^{-/-} \rightarrow Ldlr^{-/-}$ controls on HCHF diet (Figure 8). These results indicate that the relatively modest change in lipoprotein profiles (increased chylomicrons and decreased HDL) is insufficient to explain the striking increase in atherosclerosis generated by the $Lrp1^{Y63F}$ mutation. Instead, a cell autonomous effect of Lrp1 tyrosine phosphorylation in macrophages is primarily responsible for increased atherosclerosis.

## $Lrp1^{Y63F}$ promotes macrophage lipid accumulation in response to oxLDL

Histological analysis of the atherosclerotic plaque suggests a role for macrophage apoptosis as well as intracellular lipid accumulation. To examine whether the $Lrp1^{Y63F}$ mutation impacts macrophage foam cell formation, we incubated peritoneal macrophages isolated from $Ldlr^{-/-}$ and $Lrp1^{Y63F};Ldlr^{-/-}$ mice with oxLDL. Intracellular lipid accumulation was significantly higher in cultured $Lrp1^{Y63F};Ldlr^{-/-}$ macrophages compared to control cells as shown by Oil-Red O staining (Figure 9A). To understand the etiology for increased intracellular lipid accumulation, [$^3$H]-labeled cholesterol efflux experiments in peritoneal macrophages from $Ldlr^{-/-}$ and $Lrp1^{Y63F};Ldlr^{-/-}$ mice were performed. Compared to $Ldlr^{-/-}$, macrophages from $Lrp1^{Y63F};Ldlr^{-/-}$ mice had significantly reduced cholesterol efflux capacity to ApoAI, but not to HDL (Figure 9B). We next examined the expression profiles of lipid transporters involved in cholesterol transport (Figure 9C). In response to oxLDL, Cd36 was increased in $Ldlr^{-/-}$ and $Lrp1^{Y63F};Ldlr^{-/-}$ macrophages. However, there was decreased expression of the reverse cholesterol transporter Abca1 both at baseline and in response to oxLDL in $Lrp1^{Y63F};Ldlr^{-/-}$ macrophages, consistent with the cholesterol efflux data. No difference in the expression levels of Abcg1 and Srb1 were observed in the two genotypes after oxLDL stimulation. As both ABCA1 and LRP1 are transmembrane proteins, we used surface biotinylation to investigate the distribution of these proteins. Analysis of surface and total expression of Abca1 and Lrp1 in oxLDL-stimulated macrophages revealed no difference in the ratio of surface/total Abca1 or Lrp1 between $Lrp1^{Y63F};Ldlr^{-/-}$ and $Ldlr^{-/-}$ macrophages (Figure 9D), suggesting that translocation of Abca1 and Lrp1 to the cell membrane is functioning normally. Finally, cross-sectional analysis of the aortic root confirmed greatly decreased expression of Abca1 in atherosclerotic plaques from HCHF diet fed $Lrp1^{Y63F};Ldlr^{-/-}$ mice in vivo (Figure 10). Together, these results demonstrate that macrophage cholesterol efflux through Abca1 is selectively impaired by the $Lrp1^{Y63F}$ mutation.

## Role of PPARγ/LXR signaling in oxLDL-induced ABCA1 expression

We next explored transcriptional mechanisms that might be regulated by LRP1 tyrosine phosphorylation and impact ABCA1 expression. A major regulator of ABCA1 expression is LXR: both LXRα and LXRβ respond to oxidized sterols and regulate cholesterol transport (Janowski et al., 1996). In macrophages, activation of PPARγ induces LXRα transcriptional activation, thereby increasing ABCA1 expression (Chawla et al., 2001). We therefore asked whether LRP1 may regulate PPARγ/LXR. We have previously shown in SMC that LRP1 indirectly regulates LXR-mediated ABCA1 expression independent of LXR gene transcription (Zhou et al., 2009a). An additional precedent is provided by a recent study demonstrating that the LRP1 cytoplasmic domain itself can regulate gene expression by directly interacting with PPARγ (Mao et al., 2017). To investigate whether phosphorylation of LRP1 at Y63 affects the PPARγ/LXR pathway in oxLDL-induced ABCA1 expression, we tested the

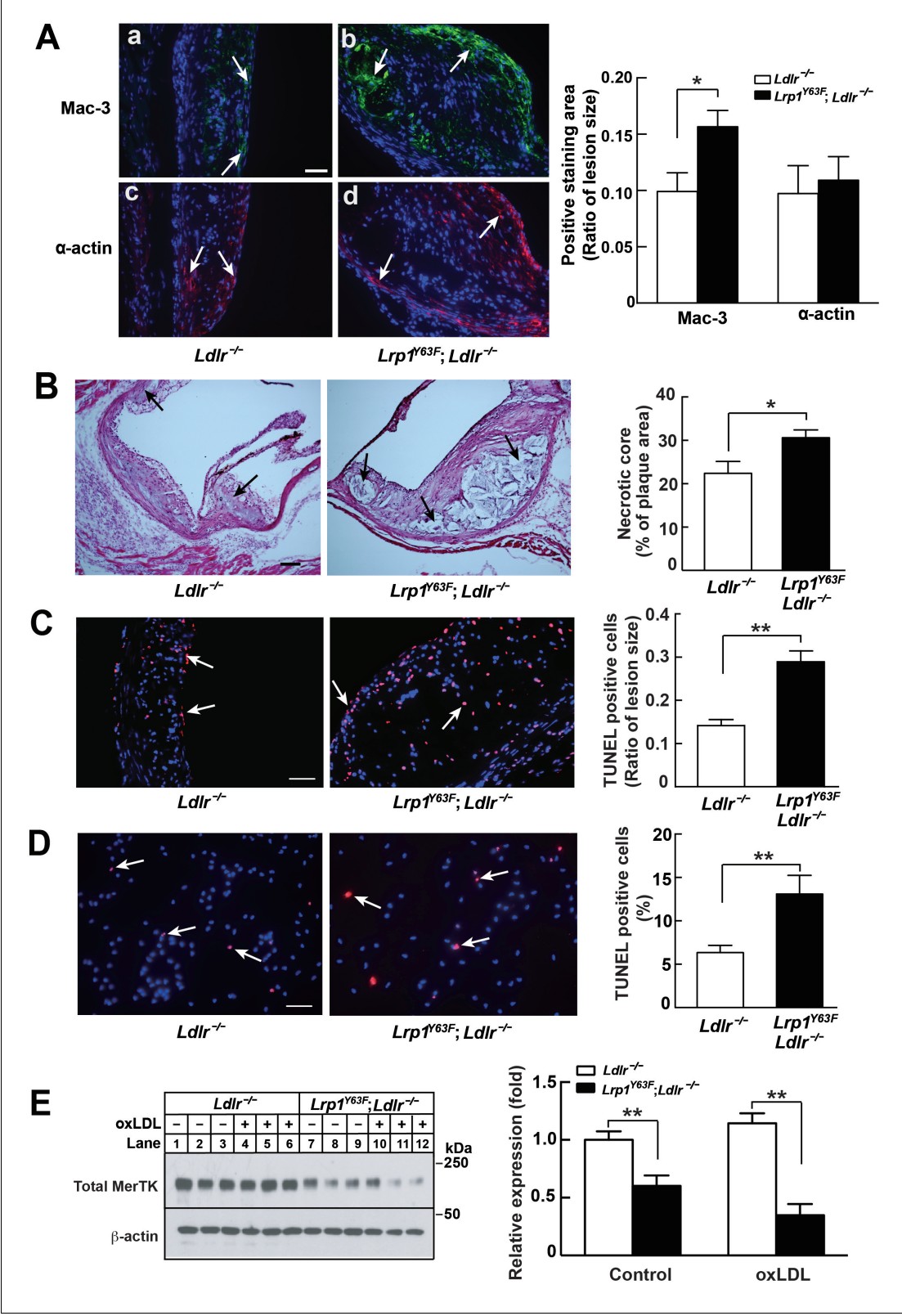

**Figure 5.** Enhanced necrotic core in atherosclerotic lesions in *Lrp1^{Y63F}*;*Ldlr^{-/-}* mice. (**A**) Left: Immunohistochemistry of aortic roots sections stained with Mac-3 (a and b, scale bar = 50 μM) and α-actin (c and d, scale bar = 50 μM). White arrows indicate positive immunofluorescent staining. Right: postive staining area was quantified. (**B**) Representative sections of H&E stained aortic roots from *Ldlr^{-/-}* (n = 13) and *Lrp1^{Y63F}*;*Ldlr^{-/-}* (n = 12) mice after 16 weeks of HCHF diet. Black arrows indicate the necrotic core (left, scale bar = 50 μM) and necrotic core area in the aortic roots was quantified (right). (**C**)
*Figure 5 continued on next page*

Figure 5 continued

Immunohistochemical analysis of apoptosis in atherosclerotic lesions in mice described above. Representative sections of TUNEL staining of the aortic roots from indicated mice (left: scale bar = 50 μM) and TUNEL-positive cells in the lesions were counted (right). (D) Immunohistochemical analysis of apoptosis in oxLDL-treated peritoneal macrophages in vitro. Representative images of TUNEL-stained macrophages with indicated genotypes (left: scale bar = 50 μM) and TUNEL-positive cells were counted (right). (E) Western blot analysis of MerTK in macrophages in the absence or presence of oxLDL. All data are mean ±SEM. *p<0.05, **p<0.01.

DOI: https://doi.org/10.7554/eLife.29292.007

transcription levels of LXR-dependent gene regulation. The $Lrp1^{Y63F}$ mutation diminished oxLDL-induced $Abca1$ gene transcription, but not $Abca7$ gene transcription, which by contrast is increased in macrophages lacking LRP1 completely (*Yancey et al., 2010*). Furthermore, the $Lrp1^{Y63F}$ mutation had no effect on $Nr1h3$ (a.k.a. $Lxr\alpha$), $Nr2h2$ (a.k.a. $Lxr\beta$), or $Pparg$ gene expression in macrophages in the presence or absence of oxLDL (*Figure 11A*), and there was no change in LXRα or PPARγ protein expression (data not shown). Using the PPARγ inhibitor T0070907, we confirmed that PPARγ activity was necessary for Abca1 expression in both $Ldlr^{-/-}$ and $Lrp1^{Y63F};Ldlr^{-/-}$ macrophages (*Figure 11B*). To further investigate the role of PPARγ/LXR activation, isolated peritoneal macrophages were incubated with the PPARγ agonist Rosiglitazone (*Figure 11C*), the nonspecific LXR agonist T0901317 (*Figure 11D*), or LXR623 (a LXRβ full agonist and LXRα partial agonist) (*Figure 11E*). Neither LXR or PPARγ agonists, over a range of doses, were able to fully correct the blunted Abca1 levels in the $Lrp1^{Y63F};Ldlr^{-/-}$ mice (*Figure 11C–E*). Together these data demonstrate that the entire PPARγ/LXR/ABCA1 axis is severely impaired by the $Lrp1^{Y63F}$ mutation in macrophages and suggest that a membrane proximal event that is dependent on Lrp1 tyrosine phosphorylation is required to activate Abca1 transcription through PPARγ and LXR.

## PI3K/AKT signaling is impaired in oxLDL-stimulated Lrp1$^{Y63F}$;Ldlr$^{-/-}$ macrophages

We have previously shown in SMCs that LRP1 regulates mitogenic signaling, leading to ERK and AKT phosphorylation (*Boucher et al., 2003*; *Zhou et al., 2009b*). Demers et al. showed that PI3K/AKT activation can modulate the PPARγ/LXR/ABCA1 axis (*Demers et al., 2009*), thereby raising the possibility that PI3K activation might be dependent upon LRP1. We thus investigated the effect of oxLDL and Lrp1 phosphorylation on PI3K/Akt signaling in macrophages. Phospho-Akt levels were significantly decreased in $Lrp1^{Y63F};Ldlr^{-/-}$ macrophages, both at baseline and in response to oxLDL (*Figure 12A*). Moreover, inhibition of the PI3K/Akt pathway with LY294002, a PI3K inhibitor, or an Akt inhibitor, in $Ldlr^{-/-}$ macrophages significantly attenuated Abca1 induction in response to oxLDL, such that Abca1 expression levels were comparable to those found in $Lrp1^{Y63F};Ldlr^{-/-}$ macrophages (*Figure 12B*). To show that PI3K/Akt is required for full induction of Abca1 expression by PPARγ/LXR, $Ldlr^{-/-}$ macrophages were pre-incubated with PI3K/Akt inhibitors followed by PPARγ/LXR agonists. Inhibition of PI3K/Akt results in incomplete induction of Abca1 by both PPARγ and LXR agonists, similar to what is observed in macrophages carrying the $Lrp1^{Y63F}$ mutation (*Figure 12C and D*). This suggested the existence of an endogenous mechanism that is dependent upon LRP1 tyrosine phosphorylation to activate PI3K/Akt and subsequently PPARγ/LXR-induced Abca1 expression.

## Knockdown of Shc1 inhibits oxLDL-induced Akt phosphorylation and Abca1 upregulation

To identify the missing link that connects LRP1 to PI3K/Akt, we evaluated the numerous adaptor proteins that interact with LRP1 (*Barnes et al., 2001*; *Gotthardt et al., 2000*). SHC1 is such an intracellular adaptor protein with an established role in mitogenic signal transduction (*Pelicci et al., 1992*). Tyrosine phosphorylation of Lrp1 is required for the binding of the intracellular adapter protein Shc1 in SMC, and the $Lrp1^{Y63F}$ mutation prevents binding of Shc1 to Lrp1 (*Figure 1*). Using co-immunoprecipitation assays, we confirmed that the interaction between phosphorylated Lrp1 and Shc1 also occurs in macrophages (*Figure 13A*). To investigate the role of Shc1 in oxLDL-induced Abca1 expression, we employed shRNA-directed gene silencing to specifically reduce the expression levels of Shc1 in $Ldlr^{-/-}$ and $Lrp1^{Y63F};Ldlr^{-/-}$ macrophages. The induction of Akt phosphorylation and Abca1 expression in response to oxLDL was significantly impaired by the knockdown of Shc1 in

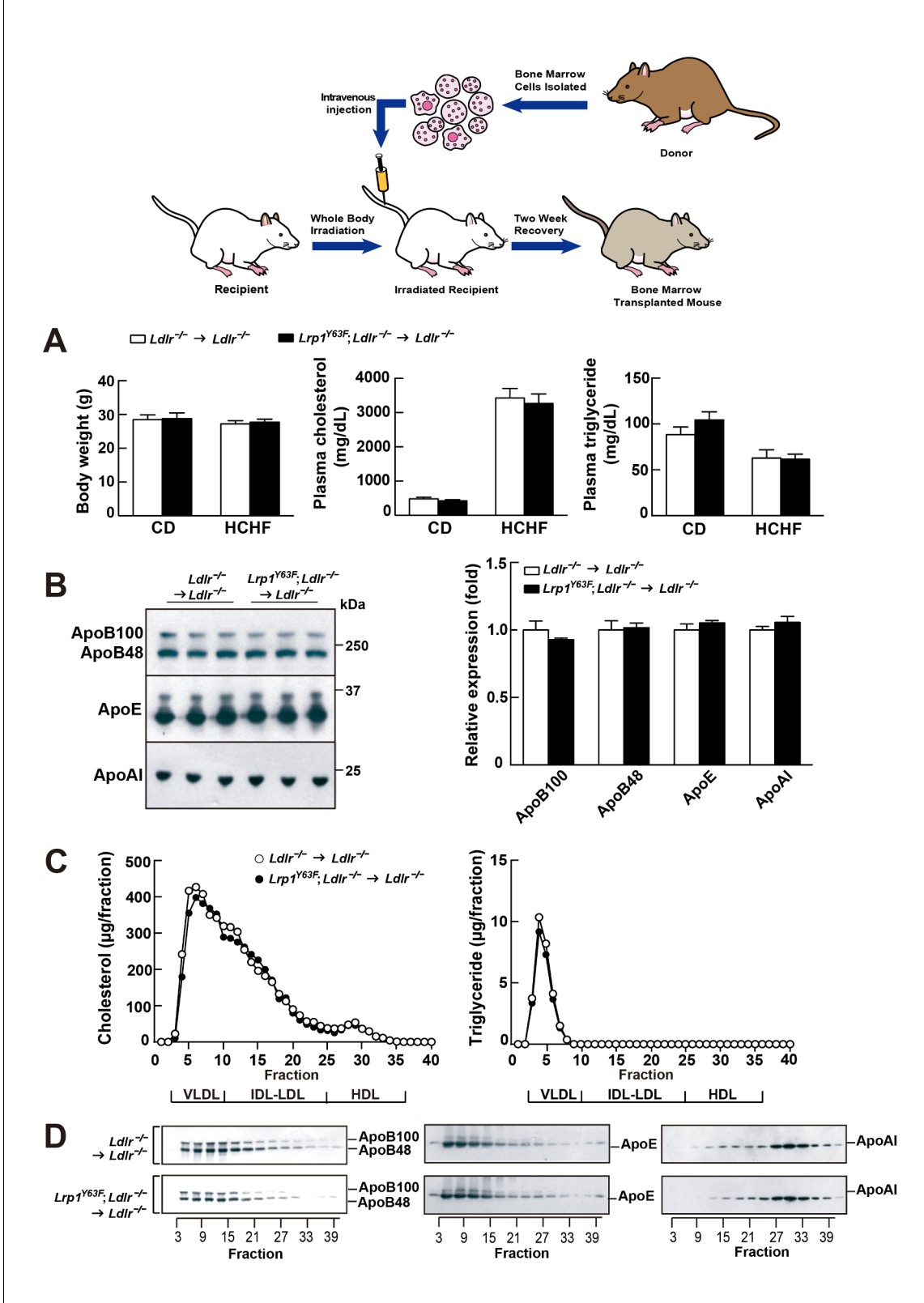

**Figure 6.** Bone marrow transplantation (BMT) with *Lrp1^{Y63F}* bone marrow does not affect lipid profiles in *Ldlr^{−/−}* mice. Diagram of bone marrow transplant is shown on the top. Bone marrows from *Lrp1^{Y63F};Ldlr^{−/−}* mice and *Ldlr^{−/−}* control mice were transplanted into γ-irradiated-*Ldlr^{−/−}* mice. After BMT, recipient mice were placed on the indicated diets for 16 weeks. (**A**) Body weight, plasma cholesterol and triglyceride levels of *Ldlr^{−/−}* → *Ldlr^{−/−}* (n = 9) and *Lrp1^{Y63F};Ldlr^{−/−}*→*Ldlr^{−/−}* (n = 15) mice on CD or HCHF diet. (**B**) Representative immunoblots and quantitative analysis of plasma

*Figure 6 continued on next page*

*Figure 6 continued*

apolipoproteins in BMT mice on 16 week HCHF diet (n = 5, each group). (**C**) FPLC plasma lipid profiles. Cholesterol (left) and triglyceride (right) in fractionated pooled plasma from HCHF-fed mice (n = 8–10, each group). (**D**) Representative Western blots of ApoB (left), ApoE (middle) and ApoAI (right) in different fractions. All data are mean ±SEM.

DOI: https://doi.org/10.7554/eLife.29292.008

*Ldlr*$^{-/-}$ macrophages, whereas Shc1 knock-down had no effect on Akt phosphorylation and Abca1 expression in *Lrp1*$^{Y63F}$;*Ldlr*$^{-/-}$ macrophages (*Figure 13B*). We therefore concluded that Akt phosphorylation and Abca1 expression is dependent on the interaction between Shc1 and phosphorylated Lrp1.

Collectively, our data provide a novel mechanistic basis for the role of macrophage LRP1 in atherogenesis. As summarized in *Figure 14A*, tyrosine phosphorylation of LRP1, which is mediated by activated Src family tyrosine kinases in a range of inflammatory or proliferative conditions (*van der Geer, 2002*), is required for interaction with the adaptor protein SHC1, which in turn augments PI3K/AKT activation. Phospho-AKT modulates PPARγ/LXR driven gene expression of which *ABCA1* is a prominent target with well-documented roles in lipid metabolism and atherosclerosis (*Aiello et al., 2002*; *Huang et al., 2015*; *Rader et al., 2009*; *Shao et al., 2014*; *Singaraja et al., 2002*; *Su et al., 2005*; *Tall and Yvan-Charvet, 2015*; *Yvan-Charvet et al., 2007*; *Zhao et al., 2010*). The activation of this LRP1/SHC1/PI3K/AKT/PPARγ/LXR axis is necessary for the full effect of nuclear hormone receptor ligand-induced and PPARγ/LXR-dependent expression of ABCA1 and the engulfment receptor MerTK. In the absence of LRP1/SHC1 interaction, the decreased expression of

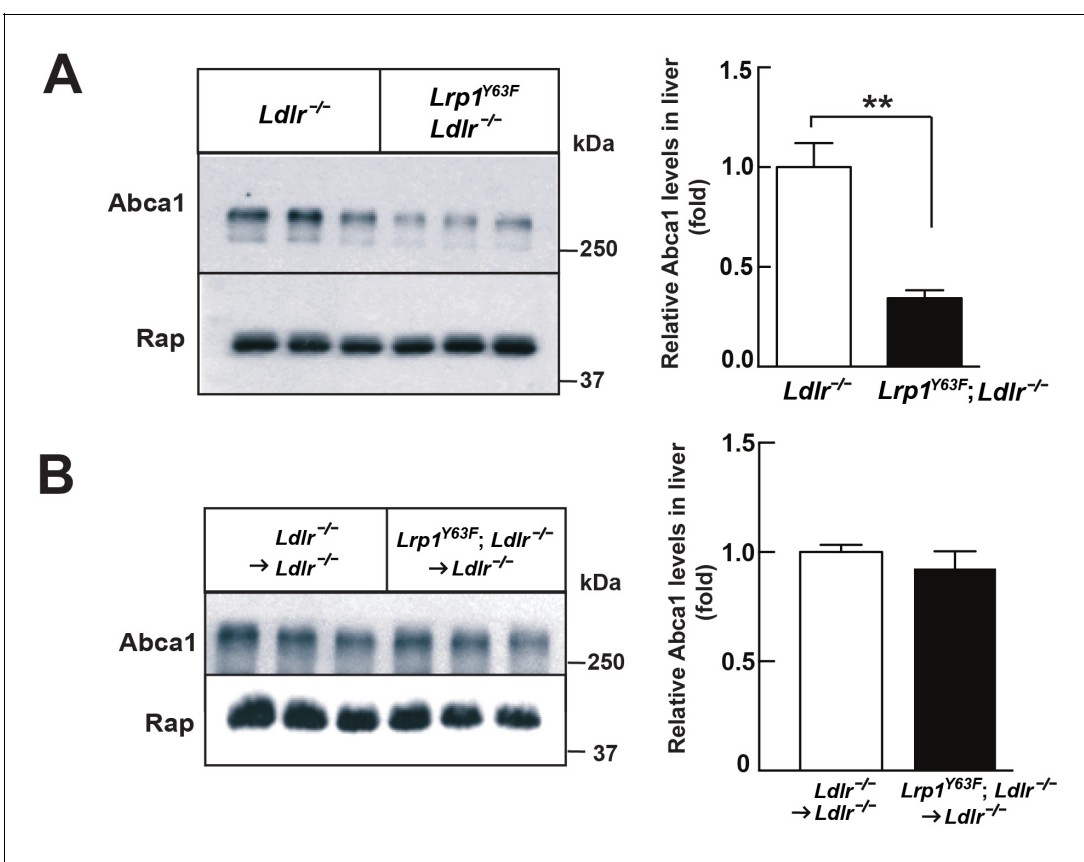

**Figure 7.** *Lrp1*$^{Y63F}$ mutation downregulates hepatic Abca1 levels. (**A**) Western blot analysis of Abca1 expression in the liver from *Lrp1*$^{Y63F}$;*Ldlr*$^{-/-}$ and *Ldlr*$^{-/-}$ mice fed with HCHF diet for 16 weeks (n = 5, each group). (**B**) Western blot analysis of Abca1 expression in the liver from 16 week HCHF-fed *Lrp1*$^{Y63F}$;*Ldlr*$^{-/-}$ and *Ldlr*$^{-/-}$ mice after BMT (n = 5, each group). All data are mean ±SEM. **p<0.01.

DOI: https://doi.org/10.7554/eLife.29292.009

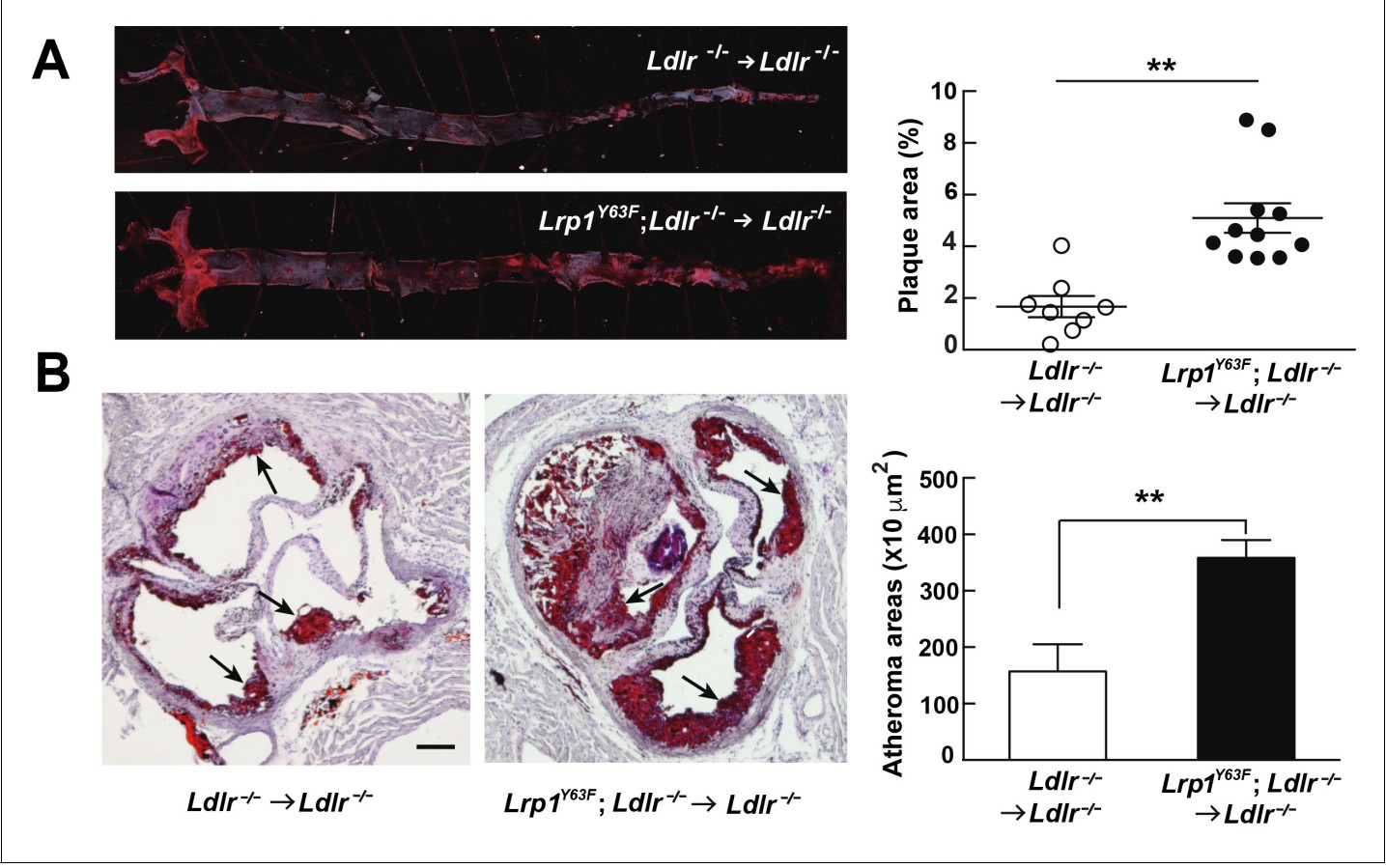

**Figure 8.** Bone marrow-derived *Lrp1*[Y63F] promotes atherosclerosis. (**A**) Representative images of en face assessment of atherosclerotic lesions in 16 week HCHF-fed recipient mice with indicated bone marrow donor mice (left) and quantification of plaque areas (right). (**B**) Representative Oil red O-stained sections of aortic roots in each BMT group (left) and quantification of atheroma areas (right). Black arrows indicate atherosclerotic plaque lesions. All data are mean ±SEM. **p<0.01.

DOI: https://doi.org/10.7554/eLife.29292.010

atheroprotective LXR target genes in macrophages, as exemplified here by *ABCA1* and *MerTK*, leads to increased intracellular lipid accumulation, macrophage foam cell formation, impaired apoptotic cell clearance and thus increased atherosclerotic plaque formation, independent of systemic lipid levels.

## Discussion

We have generated a novel knock-in mouse model to investigate the physiological role of tyrosine phosphorylation at the distal NPxY motif within the cytoplasmic domain of LRP1 and its effect on staving off atherosclerosis. Using bone marrow transplantation, we showed that the accelerated atherosclerosis in *Lrp1*[Y63F];*Ldlr*[−/−] mice is caused by a malfunction of macrophages leading to increased intracellular lipid accumulation independent of systemic lipid levels, and deceased apoptotic cell clearance. We further showed that LRP1, through a phosphotyrosine-dependent interaction with SHC1 and subsequent activation of PI3K/AKT, is required for the efficient stimulation of the anti-inflammatory and cholesterol export-promoting PPARγ and LXR pathway. In this study we have concentrated on ABCA1, a prominent, though by no means exclusive, target of this novel atheroprotective LRP1/SHC1/PI3K/AKT/PPARγ/LXR axis, which works in aggregate with other LRP1-dependent mechanisms (*Barnes et al., 2001*; *Boucher et al., 2002*; *El Asmar et al., 2016*; *Subramanian et al., 2014*; *Zhou et al., 2009a*; *Zurhove et al., 2008*) to prevent macrophage foam cell formation, suppress inflammation and promote apoptotic cell clearance (*Tall and Yvan-Charvet, 2015*).

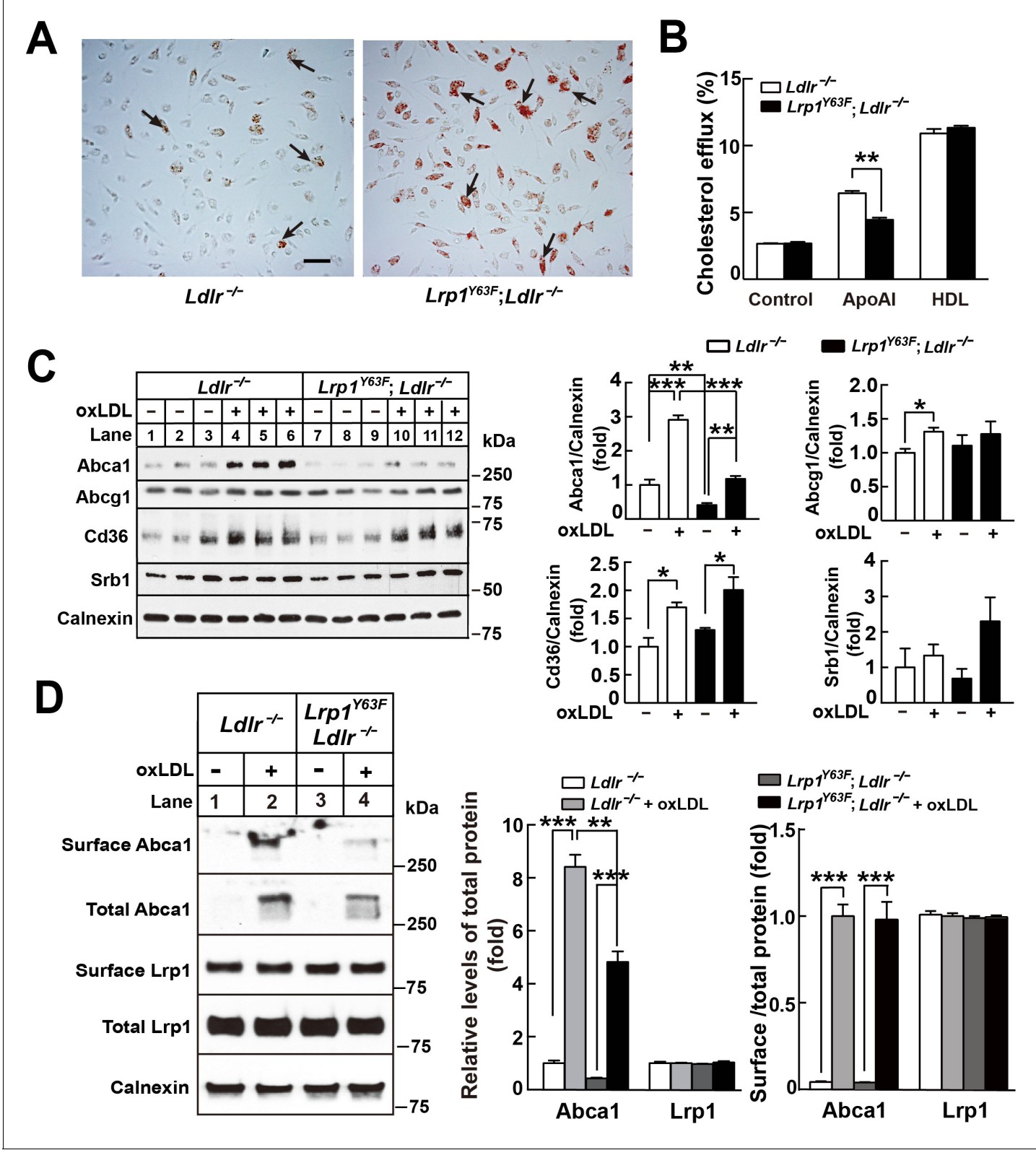

**Figure 9.** *Lrp1*[Y63F] impairs Abca1-mediated cholesterol efflux and increases lipid accumulation in macrophages. Peritoneal macrophages isolated from *Lrp1*[Y63F];*Ldlr*[−/−] and *Ldlr*[−/−] mice were starved overnight and treated with or without oxLDL (100 µg/mL) for 24 hr. (**A**) Representative photomicrographs of lipid accumulation in macrophages stained with Oil red O (left, *Ldlr*[−/−]; right, *Lrp1*[Y63F];*Ldlr*[−/−], scale bar = 50 µm). Black arrows indicate Oil red O-stained positive cells. (**B**) Macrophages isolated from *Ldlr*[−/−] and *Lrp1*[Y63F];*Ldlr*[−/−] mice were labeled with [3H]-cholesterol in the

*Figure 9 continued on next page*

Figure 9 continued

presence of oxLDL for 24 hr. After equilibration in 0.2% BSA overnight, cells were incubated with either 20 µg/ml apoAI or 30 µg/ml HDL for 4 hr. (C) Western blot analysis of lipid transporters in macrophages treated with or without oxLDL (left) and immunoblot quantitation (right). (D) After treatment with or without oxLDL, surface Abca1 and Lrp1 at plasma membrane in macrophages were biotinylated. The ratio of surface to total protein levels was analyzed by Western blotting (left) and quantification shown (right). All data are mean ±SEM from 3 to 6 independent experiments. *p<0.05, **p<0.01, ***p<0.001.

DOI: https://doi.org/10.7554/eLife.29292.011

The inability of Shc1 to bind to Lrp1$^{Y63F}$ resulted in decreased activation of PI3K/AKT and reduced expression of the LXR target genes *ABCA1* and *MerTK*, an established inducer of apoptotic cell clearance, in macrophages. Our data further demonstrate that this SHC1-mediated AKT phosphorylation is necessary for the LRP1 dependent induction of ABCA1. Rescue of *Lrp1$^{Y63F}$* macrophages with LXR and PPARγ agonists did not fully correct the diminished Abca1 levels, suggesting either the existence of an alternative mechanism wherein PI3K/AKT regulates ABCA1 expression independent of PPARγ/LXR (*Chen et al., 2012*), or that PI3K/AKT is a necessary upstream regulator of PPARγ/LXRα driven gene expression (*Demers et al., 2009*; *Ishikawa et al., 2013*).

The model we propose begins with LRP1 tyrosine phosphorylation, an event mediated by Src-family kinases (SFK), which are often localized to lipid rafts (*Simons and Toomre, 2000*). LRP1 associates with lipid rafts in the plasma membrane, thus creating transient microdomains that are cell specific (*Wu and Gonias, 2005*). Membrane organization into lipid rafts not only explains LRP1 tyrosine phosphorylation, but also underscores the relationship with ABCA1 to maintain the homeostatic balance of cellular membrane composition and signal transduction (*Figure 14B*). Through cellular cholesterol export ABCA1 regulates cellular plasma membrane cholesterol content and membrane structural organization, specifically the formation of lipid rafts (*Ito et al., 2015*; *Sezgin et al., 2017*; *Zhu et al., 2010*). In the setting of inflammation or cholesterol loading, increased proximity of LRP1 with activated Src family kinases in lipid rafts favors LRP1 tyrosine phosphorylation. This, in turn, activates a SHC1/PI3K/AKT/PPARγ/LXRα axis that promotes ABCA1 expression and cellular cholesterol export. The induction of ABCA1 leads to reduction of lipid raft cholesterol content (*Ito et al., 2015*) and dissociation of LRP1 from lipid rafts, thus creating a self-limiting negative feedback loop and returning homeostatic balance to the plasma membrane. In cases of impaired LRP1 phosphorylation, the dynamic interaction between LRP1 and lipid rafts may be intact, however the downstream signaling events triggered by LRP1 tyrosine phosphorylation cannot be initiated, thus impairing cholesterol homeostasis and plasma membrane composition.

LRP1 and ABCA1, however, play an even larger role by integrating cholesterol homeostasis with inflammatory signaling. By modulating membrane cholesterol content and lipid raft organization, macrophage ABCA1 regulates trafficking of Toll-like-receptor 4 (TLR4) to lipid rafts (*Zhu et al., 2010*). In lipid rafts, TLR4 forms a complex with TLR6 and oxLDL-binding CD36 (*Figure 14B and C*), which leads to recruitment of the adaptor MyD88, thus triggering inflammatory signaling (*Stewart et al., 2010*). ABCA1 itself has anti-inflammatory properties, as interaction of ApoA1 with ABCA1 activates JAK2/STAT3 (*Tang et al., 2009*). Finally, as demonstrated by Ito et al, induction of *Abca1* by LXRs inhibits inflammatory NF-κB and MAPK signaling pathways downstream of TLRs (*Ito et al., 2015*). Therefore, by modulating LXR-driven *ABCA1* gene expression through the SHC1/PI3K/AKT/PPARγ/LXRα axis, LRP1 is presenting itself as a pivotal regulator of the metabolic as well as inflammatory processes underlying atherosclerosis.

In addition to the indirect anti-inflammatory effect through ABCA1, there is also emerging data that LRP1 can directly regulate inflammatory gene transcription in macrophages (*Zurhove et al., 2008*). After cleavage of the LRP1 intracellular domain (ICD) by γ-secretase from the plasma membrane, the LRP1 ICD translocates into the nucleus and acts as a transcriptional repressor of genes for inflammatory cytokines (*Figure 14B*) (*Zurhove et al., 2008*). In an analogous manner in endothelial cells, the LRP1 ICD has been shown to directly interact with PPARγ and function as a transcriptional co-regulator of the PPARγ target gene Pdk4 (*Mao et al., 2017*). Together these findings underscore the integrative relationship between LRP1 and transcriptional regulators LXR and PPARγ and their interdependent roles in cholesterol homeostasis and inflammation.

A third critical macrophage process underlying atherogenesis is clearance of apoptotic cells, or efferocytosis (*Figure 14C*). Macrophage-specific deficiency of LRP1 is associated with increased

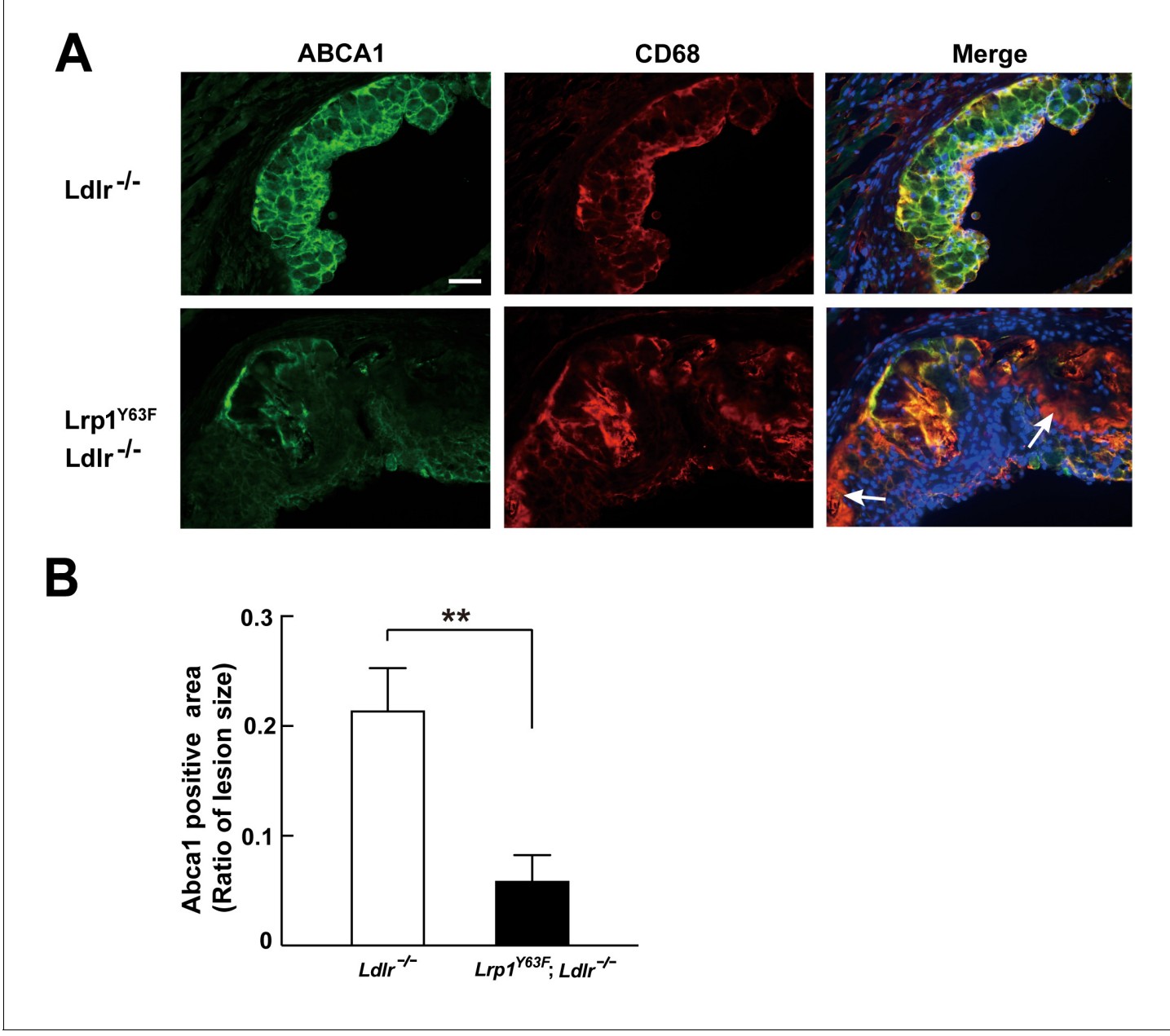

**Figure 10.** Abca1 expression is reduced in atherosclerotic lesions in *Lrp1*$^{Y63F}$;*Ldlr*$^{−/−}$ mice. (**A**) Cryo-sections of aortic roots from HCD-fed *Ldlr*$^{−/−}$ and *Lrp1*$^{Y63F}$;*Ldlr*$^{−/−}$ mice were double-stained with Abca1 (green) and Cd68 (red) antibodies. Nuclei were stained with DAPI (blue). (**B**) Abca1 immunoreactivity was determined by color and normalized to lesion size. Scale bar = 200 µM. White arrows indicate the areas of macrophages without Abca1 expression. Data are mean ±SEM from five mice for each genotype. **p<0.01.

DOI: https://doi.org/10.7554/eLife.29292.012

macrophage apoptosis and decreased efferocytosis, leading to larger necrotic cores in atherosclerotic plaque (*Misra and Pizzo, 2005*; *Yancey et al., 2010*; *Yancey et al., 2011*) Mechanistically, this has been associated with reduced levels of pAKT (*Yancey et al., 2010*), similar to the findings in our *Lrp1*$^{Y63F}$;*Ldlr*$^{−/−}$ mice.

An additional mechanism through which the *Lrp1*$^{Y63F}$ mutation may affect efferocytosis is through impaired activation of LXR. LXR-deficient macrophages have reduced phagocytic capacity (*A-Gonzalez et al., 2009*), which is associated with decreased expression of *Mertk*, an LXR-responsive gene. MerTK is critical for efficient macrophage phagocytosis and efferocytosis, and deficiencies in

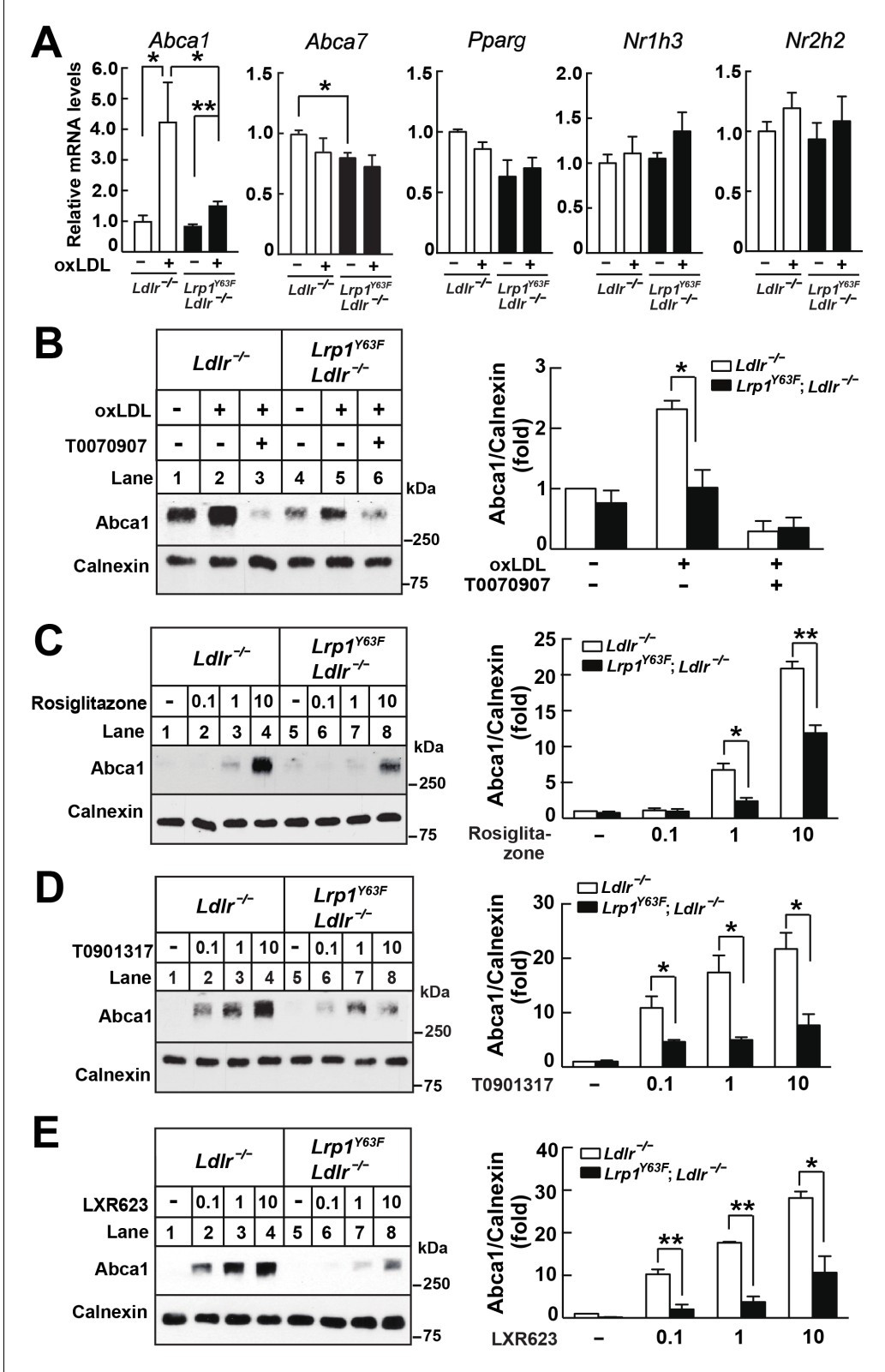

**Figure 11.** *Lrp1^{Y63F}* impairs Abca1 induction through inhibiting the PPARγ/LXR pathway. (**A**) Real-time PCR analysis of *Abca1, Abca7, Pparg, Nr1h3 and Nr2h2* expression in macrophages treated with or without ox-LDL. (**B–E**) Isolated peritoneal macrophages from *Lrp1^{Y63F};Ldlr^{−/−}* and *Ldlr^{−/−}* were pre-incubated for 1 hr with (**B**) T0070907 (PPARγ antagonist, 10 μM), (**C**) Rosiglitizone (PPARγ agonist) at indicated concentrations in μM, (**D**) T0901317 (nonspecific LXR agonist) at indicated concentrations in μM, (**E**) LXR623 (LXRβ full agonist and LXRα partial agonist) at indicated concentrations in μM,

*Figure 11 continued on next page*

*Figure 11 continued*

followed by the treatment with or without oxLDL for an additional 24 hr. Abca1 expression was assessed by immunoblot analysis (left). Protein expression was quantified and analyzed (right). Data are mean ±SEM from three independent experiments. *p<0.05; **p<0.01.

DOI: https://doi.org/10.7554/eLife.29292.013

macrophage MerTK expression contribute to increased necrotic cores in atherosclerotic lesions (*Li et al., 2006*; *Scott et al., 2001*; *Thorp et al., 2008*).

Finally, LRP1 itself is necessary for the efficient phagocytosis of apoptotic cells, as it is required for the formation of a multiprotein complex involving LRP1, the receptor tyrosine kinase AXL, and the scaffolding protein RANBP9 (*Subramanian et al., 2014*). As described by Subramanian et al., whereas AXL is involved in the recognition and binding of apoptotic cells, LRP1, by associating with GULP1 (*Su et al., 2002*), is required for the final engulfment of the dead cells. Whether these mechanisms, which are distinct from the Src kinase family-mediated activation of the LRP1/Shc1 signaling cascade, are themselves also modulated by LRP1 tyrosine phosphorylation remains currently unknown, but can now be tested using the knockin mouse model we have reported here.

Our results, which show an identical atherogenic lipoprotein profile upon cholesterol-feeding of $Ldlr^{-/-}$ mice transplanted with $Ldlr^{-/-}$ or $Lrp1^{Y63F};Ldlr^{-/-}$ bone marrow, differ on first glance from those of Bi et al. (*Bi et al., 2014*), who showed reduced plasma lipid levels in mice lacking Abca1 completely in their monocytes/macrophages. This effectively negated the proatherogenic effect of *Abca1* deficiency in lesion macrophages. In their model, however, elevated plasma cholesterol levels induced by cholesterol feeding and the resulting cholesterol overload of circulating cells led to the secretion of inflammatory cytokines, which reduced VLDL secretion by the liver and thus resulted in a less atherogenic plasma lipid profile, which subsided on a low cholesterol chow diet. In our case, monocyte/macrophages carrying the Y63F mutation in *Lrp1* retain residual, though substantially lower, Abca1 expression, which would protect the circulating cells from cholesterol overload in a hyperlipidemic, cholesterol-fed state.

In conclusion, our current studies have revealed a novel mechanism through which macrophage LRP1 not only regulates ABCA1 expression to maintain cholesterol efflux, but also integrates cellular cholesterol homeostasis with inflammation and efferocytosis. In aggregate, these macrophage-specific processes demonstrate the atheroprotective role of LRP1 in mechanisms distinct from those found in SMCs. Together, these results underscore the cell-specific nature of LRP1-mediated signaling and highlight the central role of LRP1 in integrating cellular cholesterol homeostasis with other atheroprotective processes.

## Materials and methods

### Materials

The PPARγ inhibitor T0070907 (cat# T8703), LXR activator T0901317 (cat# T2320), LXR 623 (ApexBio), Rosiglitizone (cat# R2408), LY294002 (cat# L9908), and Akt inhibitor 1,3-dihydro-1-(1-((4-(6-phenyl-1H-imidazo[4,5-g]quinoxalin-7-yl)phenyl)methyl)-4-piperidinyl-2H-bezimidazole-2-one trifluoroacetate salt hydrate (cat# A6730), and In Situ Cell Death Detection Kit, TMR red (cat# 12156792910) were purchased from Sigma-Aldrich (St. Louis, MO). EZ-LinkTM-Sulfo-NHS-SS-Biotin (cat# 21331), cholesterol kit (cat# TR13421) and triglyceride kit (cat# TR22421) were purchased from Thermo Fisher Scientific (Rockfold, IL). Antibodies against ABCG1 (cat# ab36969), β-actin (cat# ab8227), CD68 (cat# ab955), α-actin (cat# ab5694), and LXRα (cat# ab41902) were purchased from Abcam (Cambridge MA). Antibodies against ABCA1 (cat# NB400-164), SR-B1 (cat#NB400-104), and CD36 (cat# NB400-144) were from Novus Biologicals (Littleton, CO). Antibodies against cPLA2 (cat# 2832), p-cPLA2 (cat# 2831), Calnexin (cat#2433), total Akt (cat# 9272 s), p-Akt (cat# 9271), PDGFRβ (cat# 3169), p-PDGFRβ (cat# 88H8), p-Smad2/3 (cat# 3108), PPARγ (cat#2443), Erk 1/2 (cat# 4695) and p-Erk1/2 (cat# 4370) were purchased from Cell Signaling (Danvers, MA). Antibodies against Shc1 (cat# 610081), Dab2 (cat# 610465), and Mac-3 (cat# 550292) were purchased from BD Transduction Laboratories (Franklin Lakes, NJ). The antibodies against apoB (cat# 178467), apoE (cat# 178479), PDGF-BB (cat# GF149), and phosphotyrosine 4G10 antibody (cat# 05–1050) and were purchased from EMD Millipore (Billerica, MA). The antibody against MerTK (cat# AF591) was purchased from R

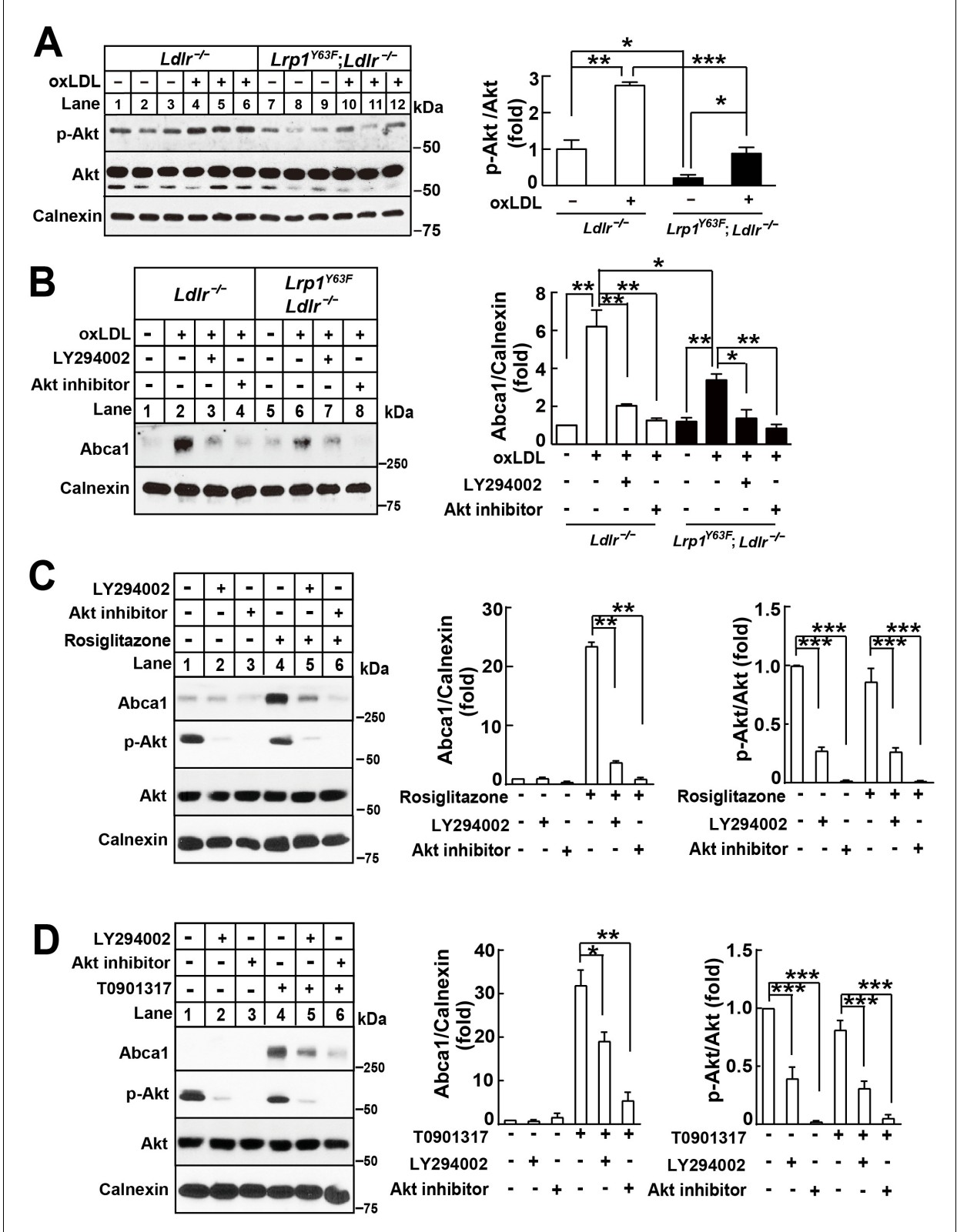

**Figure 12.** *Lrp1*$^{Y63F}$-induced impairment of Abca1 up-regulation is dependent on PI3K/Akt activities. (**A**) Western blot analysis of Akt phosphorylation in macrophages treated with or without ox-LDL. (**B**) Isolated peritoneal macrophages from *Lrp1*$^{Y63F}$;*Ldlr*$^{-/-}$ and *Ldlr*$^{-/-}$ were pre-incubated with LY294002 (PI3K antagonist, 10 μM) or Akt inhibitor (10 μM) for 1 hr, followed by ox-LDL treatment for an additional 24 hr. Abca1 protein levels were analyzed by Western blot analysis (left) and quantification is shown (right). (**C–D**) Macrophages were pretreated with LY294002 (10 μM) or Akt inhibitor (10 μM) for 1

*Figure 12 continued on next page*

*Figure 12 continued*

hr, followed by the application of 10 µM rosiglitizone (**C**) or 10 µM T0901317 (**D**) for an additional 24 hr. Abca1 and pAkt expression were assessed by immunoblot analysis (left) and quantification is shown (right). Data are mean ±SEM from three independent experiments. *p<0.05; **p<0.01; ***p<0.001.

DOI: https://doi.org/10.7554/eLife.29292.014

and D Systems. Purified human apoAI was a gift from Mingxia Liu and John S. Parks at Wake Forest School of Medicine. A rabbit polyclonal antibody against a C-terminal epitope of LRP1 was synthesized as previously described (*Herz et al., 1988*).

## Animals

The targeting strategy used to generate the mice carrying the $Lrp1^{Y63F}$ substitution is illustrated in *Figure 1* and detailed below. 8 week old mice were fed normal rodent Chow diet (Harlan Laboratories, Indianapolis, IN) or a high cholesterol/high fat (HCHF) diet containing 21% (w/w) milk fat, 1.25% (w/w) cholesterol and 0.5% (w/w) cholic acid (TD 02028, Harlan Laboratories, Indianapolis, IN) for 16 weeks.

## Generation of $Lrp1^{Y63F}$;$Ldlr^{-/-}$ mice

A loxP-flanked exon 88-exon 89 fragment followed by exon 88-neomycin-exon 89 cassette was inserted downstream of exon 87 of the murine LRP1 gene (*Figure 1*). The TAT sequence in the distal NPxY motif of the second exon 89 encoding for tyrosine 63 was mutated to TTT coding for phenylalanine by site-directed mutagenesis. The construct was electroporated into murine 129Sv/J embryonic stem cells. Homologous recombinant clones were identified by Southern blot analysis. A positive ES cell clone was injected into blastocysts to generate germ line chimaeras. Mutant offspring were generated by inter-crossing with Meox-Cre transgenic mice to remove the first exons of 88 and 89. Homozygous $Lrp1^{Y63F}$ knock-in mutant mice ($Lrp1^{ki/ki}$, $Lrp1^{Y63F}$) were subsequently crossed to homozygous LDL receptor knockout mice ($Ldlr^{-/-}$) on a C57Bl/6J background. The resultant $Lrp1^{Y63F}$;$Ldlr^{-/-}$ and $Lrp1^{loxp/loxp}Ldlr^{-/-}$ (=$Ldlr^{-/-}$) and their offspring were used for further experimental analysis.

## Bone marrow transplant

Bone marrow recipients ($Ldlr^{-/-}$ mice) were treated twice with 500 rad with a 4 hr break between exposures. Bone marrow was flushed from the femurs, tibia and fibulas of donor mice ($Ldlr^{-/-}$ or $Lrp1^{Y63F}$;$Ldlr^{-/-}$ mice) using PBS with 2% FCS. The bone marrow suspension was passed three times through an 18 g needle to create a single cell suspension and filtered on a 100 uM cell filter. Suspension was centrifuged at 1000 g for 6 min then diluted to $2 \times 10^7$ cells/ml in low glucose DMEM. Recipient mice were anesthetized using isoflurane and received a 100 ul injection of the cell dilution via the retro-orbital plexus. Post-transplant recipient mice were housed in sterile caging and administered prophylactic antibiotics in their drinking water for two weeks, followed by the indicated diet for 16 weeks.

## Primary cell derivation and culture

Primary mouse aortic smooth muscle cells were explanted from the thoracic aorta of 8–10 week old mice with indicated genotype, following the protocol by *Clowes et al. (1994)*. Briefly, the aortas were dissected under sterile conditions and the connective tissue and adventitia were removed carefully. The aortas were opened longitudinally and the intimas were scraped on luminal surface. Then the aortas were minced into small pieces and placed into a T25 flask with high glucose (4.5 g/L) DMEM containing 15% FCS, 100 U/ml penicillin, 100 mg/ml streptomycin, 20 mM L-glutamine. Both explants and cells were cultured at 37°C in 5% $CO_2$. Cells were detached by incubation with 0.25% trypsin-EDTA solution. Cell of passages 4–6 were used for the experiments under the indicated conditions.

Mouse peritoneal macrophages from $Ldlr^{-/-}$ and $Lrp1^{Y63F}$;$Ldlr^{-/-}$ mice were isolated 3 days after intraperitoneal injection of 3% thioglycollate solution and cultured in RPMI-1640 Medium containing 10% FBS. After overnight starvation, macrophages were treated under the indicated conditions,

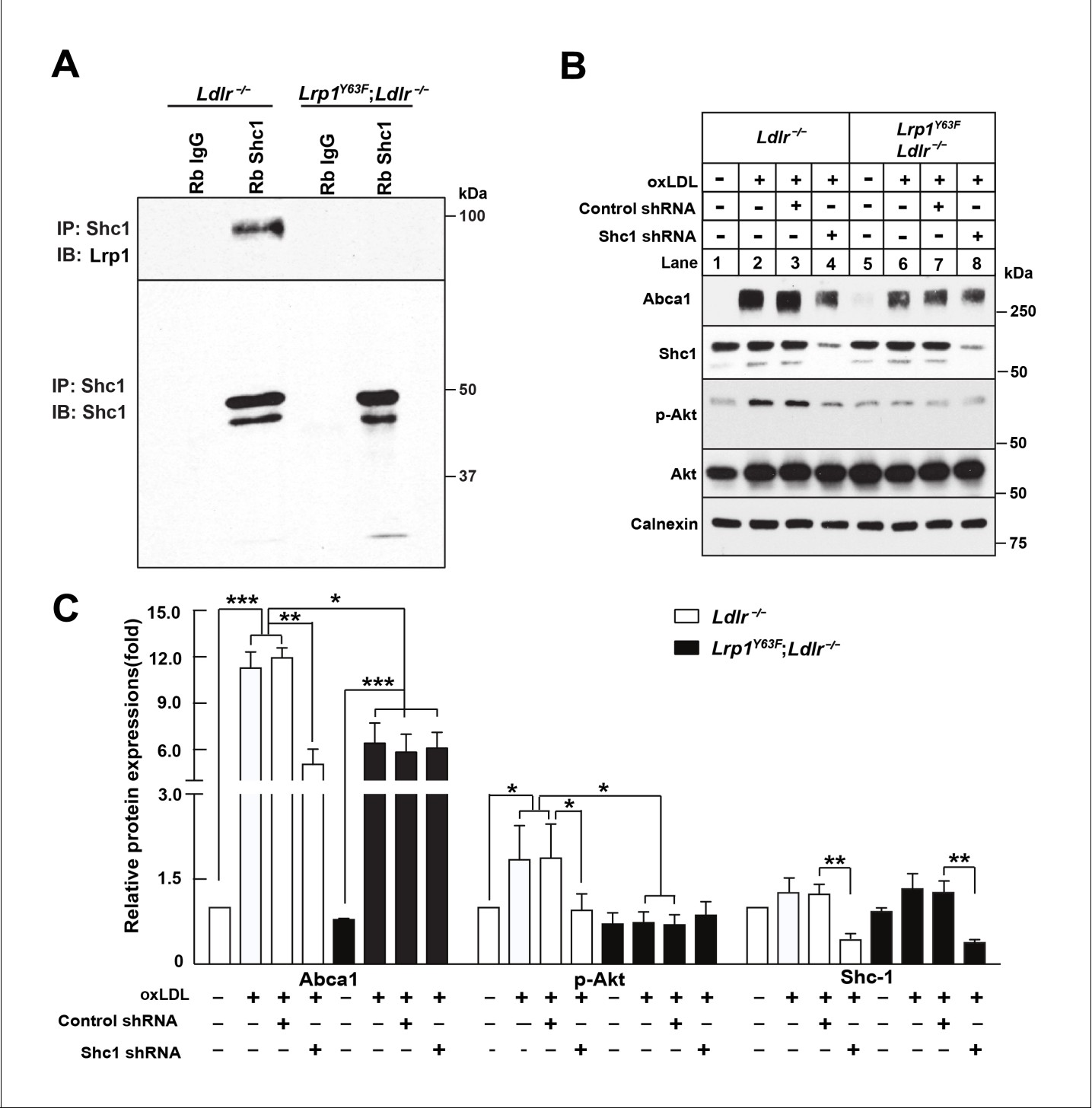

**Figure 13.** Binding of LRP1 to Shc1 is required for oxLDL-induced Abca1 up-regulation in macrophages. (**A**) Representative immunoblots probed for LRP1 (upper) and total immunoprecipitated Shc1 levels in input cell lysate (lower) from co-immunoprecipitation assays using antibodies against Shc1 (Rb Shc1) or rabbit IgG (Rb IgG). Macrophages were treated with oxLDL for 24 hr prior to immunoprecipitation. (**B**) Peritoneal macrophages were infected with lentivirus expressing scramble shRNA or Shc1 shRNA for 48 hr to knockdown endogenous Shc1 expression, followed by oxLDL treatment for an additional 24 hr. Protein expression was determined by quantitative Western blot analysis. from three independent experiments (**C**). *p<0.05; **p<0.01; ***p<0.001.

DOI: https://doi.org/10.7554/eLife.29292.015

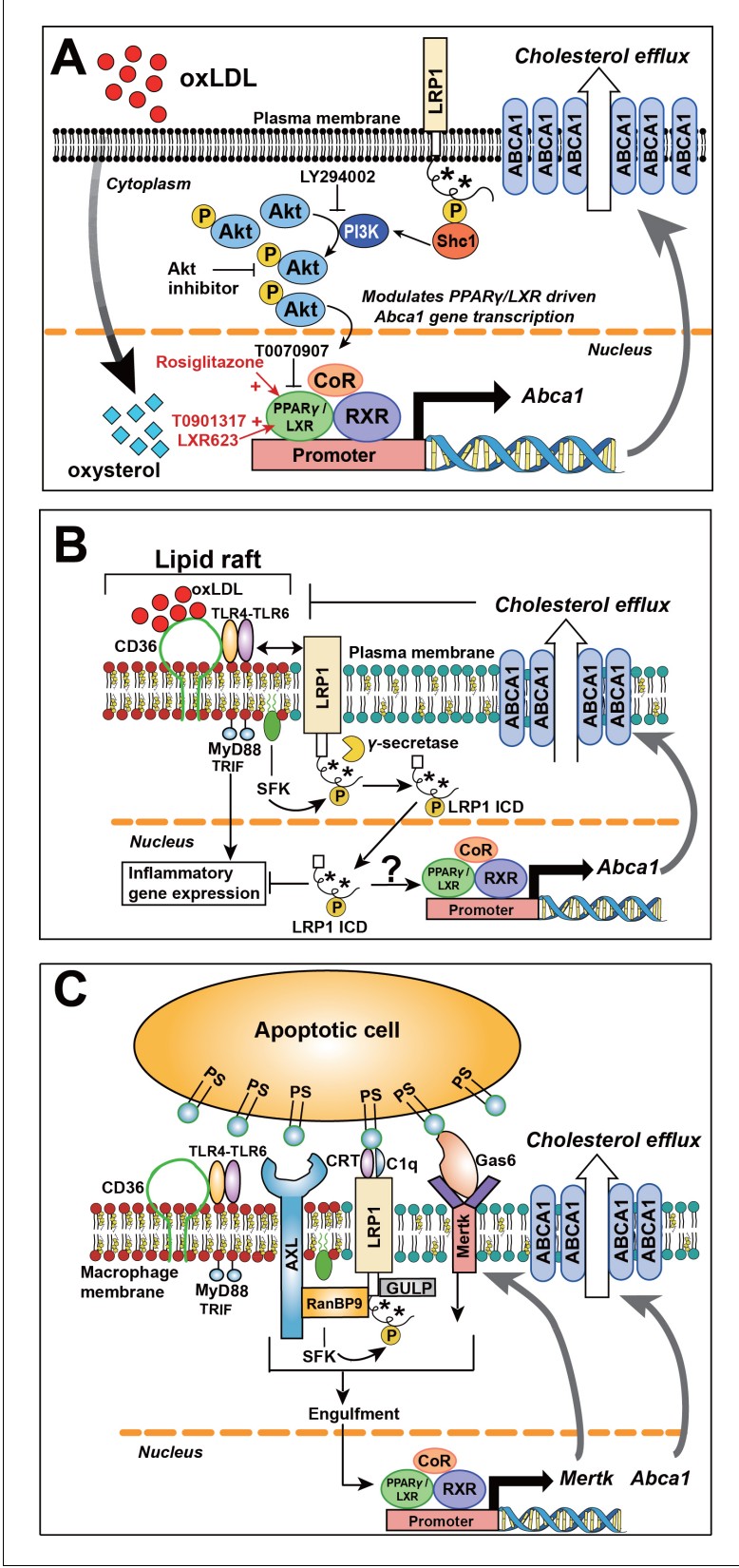

**Figure 14.** Role of LRP1 NPxY phosphorylation in macrophage lipid accumulation and atherogenesis. (**A**) Tyrosine phosphorylation of LRP1 NPxY motifs (as indicated by asterisks), specifically the distal NPxY motif, is necessary for

*Figure 14 continued on next page*

*Figure 14 continued*

interaction with Shc1, which in turn activates PI3K/Akt signaling. Phospho-Akt modulates PPARγ/LXR driven *Abca1* gene transcription and is necessary for the full effect of ox-LDL and PPARγ/LXR induced expression of Abca1 to mediate cholesterol efflux. Inhibition of PI3K/Akt (using LY294002 or Akt inhibitor) results in only partial induction of *Abca*1 in response to ox-LDL and PPARγ/LXR agonists (rosiglitazone, T0901317, LXR623). (**B**) LRP1 integrates inflammatory signals and cholesterol homeostasis in macrophages. Inflammatory signals are initiated by the interaction between oxLDL and CD36/TLR4-6 in lipid rafts. In the setting of inflammation or cholesterol loading, increased proximity of LRP1 with activated SFKs in lipid rafts favors LRP1 tyrosine phosphorylation. This, in turn, activates a Shc1/PI3K/Akt/Pparγ/Lxrα axis that promotes Abca1 expression and cellular cholesterol export, which then leads to reduction of lipid raft cholesterol content and dissociation of LRP1 from the lipid raft, thus creating a negative feedback loop. LRP1 can also be cleaved by γ–secretase, thereby releasing the intracellular domain (ICD) which translocates to the nucleus where it suppresses inflammatory gene expression (*Zurhove et al., 2008*). Whether the LRP1 ICD, in a phosphorylated or unphosphorylated state, can also alter ABCA1 expression is currently unknown. CD36, cluster of differentiation 36; TLR, toll like receptor; MyD88, myeloid differentiation primary response protein 88; TRIF, TIR domain-containing adaptor protein inducing IFNβ; SFK, SRC family kinase. (**C**) LRP1 regulates clearance of apoptotic cells (AC) through efferocytosis. LRP1 coordinates with other receptors, such AXL and MerTK, and adaptors such as RanBP9 and GULP, to recognize and engulf apoptotic cells. Engulfment of apoptotic cells is a critical step in the activation of LXR-responsive genes, including *Abca1* and *Mertk*. PS, protein S; CRT, cell surface calreticulin; GAS6, growth arrest-specific 6; AXL, receptor tyrosine kinase; MerTK, Proto-oncogene tyrosine-protein kinase; RANBP9, RAN binding protein 9; GULP, engulfment adaptor PTB domain containing one.

DOI: https://doi.org/10.7554/eLife.29292.016

followed by Oil Red O staining or lysis for western blot analysis. Oxidized LDL (oxLDL) was prepared from LDL isolated from human plasma, which was treated with 10 μM $CuSO_4$ for 24 hr at 37°C and then dialyzed in 0.09% saline containing 0.24 mM EDTA.

## Smooth muscle cell migration and proliferation assays

SMC migration was assayed using a 6.5 mm- Transwell with 8 μm polycarbonate pore membrane inserts (cat# 3422, Corning). $5 \times 10^4$ cells in 100 μl were loaded into the top chamber of each well while the lower chambers were filled with 0.5% DMEM with or without PDGF-BB (10 ng/ml). After incubating at 37°C in 5% $CO_2$ for 6 hr, non-migrated cells were scraped from the upper surface of the membrane insert. Cells on the lower surface were fixed with methanol and stained with Harris Modified Hematoxylin (HHS-16, Sigma). The number of SMC on the lower surface of the membrane insert was determined by counting four continuous high-power (200×) fields of constant area per well. Experiments were performed three times in duplicate wells.

For SMC proliferation assays, 5000 cells were seeded into each well of 96-well plates and starved for 24 hr. Cells were incubated with or without 10 ng/ml PDGF-BB for 24 hr and then stained using BrdU Cell Proliferation Assay Kit (cat# 6813, Cell Signaling Technology), following the instructions.

## Atherosclerotic plaque analysis

After 16 weeks of HCHF diet feeding, mice were subjected to an overnight fasting period and were then euthanized with isoflurane. Blood samples were collected for future use. For tissue preparation, the heart and entire aorta to the iliac bifurcation were harvested after perfusion with phosphate buffered saline (PBS) followed by 4% paraformaldehyde (Electron Microscopy Sciences, Hatfield, PA). After 24 hr in 4% paraformaldehyde, tissues were stored in 30% sucrose solution in PBS until further analysis. Aortas were dissected and opened longitudinally under a dissection microscope (Model Z30L, Cambridge Instruments). Hearts were embedded in Tissue-Tek O.C.T. compound and frozen to −20°C for analysis of atherosclerotic lesions at the aortic root. Three 10 μm thick cryosections spaced 100 μm apart were used for analysis. To evaluate the lesion size, the cryosections and whole aorta were stained with Oil Red O or hematoxylin and eosin. Semi-quantitative immunostaining was performed using primary antibodies against Mac-3 (1:200; BD Pharmingen), α–smooth muscle actin (1:200; Abcam), ABCA1 (1:50; Novus Biologicals) or CD68 (1:50; Abcam) and fluorescently labeled secondary antibodies goat anti-rat Alexa Fluor 488 (Thermo Fisher Scientific, A11006), goat anti-rabbit Alexa Fluor 594 (Thermo Fisher Scientific, A11012), or goat anti-mouse Alexa Fluor 594 (Thermo Fisher Scientific, A11032). Nuclei were counterstained with DAPI (Thermo Fisher Scientific, P36935).

Images were obtained with a Zeiss Axiophot microscope, and the percentage of lesion area that was positively stained was determined using Image-Pro v.6.2 (Media Cybernetics).

## Apoptosis analysis in vivo and in vitro

Cryo-sections of aortic roots or mouse peritoneal macrophages were prepared as described above and were stained using an In Situ Cell Death Detection Kit, following the manufacture's instructions. Briefly, tissues or cells were rinsed in PBS and fixed with 4% paraformaldehyde solution at room temperature for 15 min, followed by washing in PBS for 30 min. Samples were incubated with 0.1% Triton X-100 in cold PBS for 2 min and rinsed twice in PBS, then incubated with TUNEL reaction mixture for 1 hr at 37°C in a humidified chamber in the dark. Samples were washed twice in PBS and analyzed directly by fluorescence microscopy. Nuclei were counterstained with DAPI.

## Real time PCR

Total RNA from peritoneal macrophages treated with or without oxLDL was isolated by using TRIzol Reagent (Life technologies) and cDNA was prepared with a RT-kit (Applied Biosystems). The expression levels of *Abca1*, *Pparg*, *Nr1h3* and *Nr2h2* were measured by RT-PCR on 7900HT Fast Real-time PCR system with SYBR Green reagents and the following primer sets: mouse *Abca1* (Forward: 5'-CG TTTCCGGGGAAGTGTCCTA-3', Reverse: 5'-GCTAGAGATGACAAGGAGGATGGA-3'), mouse *Abca7* (Forward: 5'- ATCCTAGTGGCTGTCCGTCA-3', Reverse: 5'-ATGGCTTGTTTGGAAAGTGG-3'), mouse *Pparg* (Forward: 5'- CACAATGCCATCAGGTTTGG-3', Reverse: 5'-GCTGGTCGATATCAC TGGAGATC-3'), mouse *Nr1h3* (Forward; 5'-TCTGGAGACGTCACGGAGGTA-3', Reverse: 5'-CCCGGTTGTAACTGAAGTCCTT-3'), *Nr2h2* (Forward: 5'-CTCCCACCCACGCTTACAC-3', Reverse: 5'-GCCCTAACCTCTCTCCACTCA-3') and mouse *Cyclophilin* (Forward: 5'-TGGAGAGCACCAAGA-CAGACA-3', Reverse: 5'-TGCCGGAGTCGACAATGAT-3'), which was used as the internal standard.

## Surface protein biotinylation

Mouse peritoneal macrophages were grown in 6-well culture dishes and cell surface proteins were biotinylated as previously described.(*Ding et al., 2016*) After ox-LDL treatment, macrophages were washed with cold PBS buffer and then incubated in PBS buffer containing 1.0 mg/ml sulfo-NHS-SS-biotin (Pierce) for 30 min at 4°C. Excess reagent was quenched by rinsing in cold PBS containing 100 mM glycine. Cell lysates were prepared in 160 µl of RIPA lysis buffer [50 mM Tris-HCl, 150 mM NaCl, 1% NP-40, 2 mM EDTA, 2 mM MgCl2, and protease inhibitor mixture (Sigma), (pH8.0)]. After 20 min incubated at 4°C, lysates were collected and centrifuged at 14,000 × rpm for 10 min. 100 µg of total proteins were incubated with 100 µl of NeutrAvidin agarose (Pierce) at 4°C for 1.5 hr. Agarose pellets were washed three times using washing buffer [500 mM NaCl, 150 mM Tris-HCl, 0.5% Triton X-100 (pH8.0)], biotinylated surface proteins were eluted from agarose beads by boiling in 2x SDS sample loading buffer. Protein were separated by sodium dodecyl sulfate polyacrylamide gel electrophoresis (SDS-PAGE), transferred to nitrocellulose, and blotted with different antibodies.

## Cholesterol efflux assay

Peritoneal macrophages were seeded in 24-well plates, labeled with 2 µCi/ml [$^3$H]cholesterol (cat# 8794, Sigma-Aldrich), and loaded with 100 µg/ml oxLDL for 24 hr. Cells were then washed and equilibrated with 0.2% BSA-DMEM medium overnight. To determine cholesterol efflux, medium containing 20 µg/ml free apoA-I or 25 µg/ml HDL was added to cells. After 4 hr, aliquots of the medium were collected, and the [$^3$H]cholesterol released was measured by liquid scintillation counting. The [$^3$H]cholesterol remaining in the cells was determined by extracting the cell lipids in 1N NaOH and measured by liquid scintillation counting. The cholesterol efflux capacity (efflux%) is expressed as medium count/(medium count +cell count) X 100%.

## Plasma analysis

Plasma samples were obtained by cardiac puncture from mice fasted overnight. Total plasma cholesterol and triglyceride levels were determined by commercially available enzymatic kits (TR13421 and TR22421, Thermo Scientific, Waltham, MA) following manufacturer's instructions. For lipoprotein profile analysis, plasma samples were prepared by using Microvette 500K3E EDTA-columns (Sarstedt, Germany) following the same procedure as described for serum. Plasma lipoproteins were

fractionated by fast-protein liquid chromatography (FPLC) using a Superose six column (10/300 GL). Cholesterol and triglyceride content of all fractions was measured by using the same method as described for serum lipid levels.

## Western blot

Briefly, 20 µg of protein extracted from specified tissues or primary cultured cells or 20 µl of fractionated plasma samples was loaded on SDS-PAGE gel (BioRad, Hercules, CA) and transferred to nitrocellulose membranes (HybondTM-C Extra. RPN303, Amersham Biosciences, Piscataway, NJ). Blots were developed, visualized, and analyzed using the LiCor Odyssey CLX or ECL.

## Co-immunoprecipitation

Primary smooth muscle cells (SMC) were cultured in serum-deprived DMEM medium (4500 mg/L glucose, Sigma-Aldrich) for 24 hr and then treated with 10 ng/ml PDGF-BB for 10 min. To show reduced phosphorylation in $Lrp1^{Y63F}$ SMC, membrane fractions were obtained. Briefly, cell pellets were homogenized in homogenization buffer (20 mM TrisCl, 2 mM $MgCl_2$, 0.25M Sucrose, 10 mM EDTA, and 10 mM EGTA) and spun down at low speed at 800 g for 5 min. Supernatants were transferred to into new tubes for ultracentrifugation at 55,000 rpm for 30 min (MX-120, Thermo Scientific). After removal of the supernatant, the pellet was re-suspended in re-suspension buffer (50 mM TrisCl, 2 mM $CaCl_2$, 80 mM NaCl, and 1% NP-40). 500 µg membrane protein extract was used for immunoprecipitation (IP) and pre-cleared with non-immune serum. Non-specific binding was precipitated with protein A DynaBeads (Invitrogen, Grand Island, NY). Supernatant was mixed with mouse anti-phospho-tyrosine antibody at 4°C for 4 hr, and precipitated with DynaBeads at 4°C overnight. The antigen-antibody-protein complex was washed three times with wash buffer (500 mM NaCl, 150 mM Tri-HCl, 0.5% Triton-X100, pH8.0). The washed antigen-antibody-protein complex was resolved on SDS-PAGE gel and immunoblotted with rabbit anti-LRP1 antibody.

To show altered adaptor-protein binding properties in the $Lrp1^{Y63F}$ knock in mutation, SMC were treated with 10 ng/ml PDGF BB for 10 min following a starvation period of 24 hr. Cells were cross-linked using 0.05 µM DSP (Thermo Scientific, Waltham, MA) for 30 min and lysed in PBS containing 1% TritonX with proteinase inhibitor cocktail and phosphatase inhibitor for 30 min. Immunoprecipitation and Western blots were performed as described for the reduced phosphorylation experiment with secondary antibodies against adaptor proteins Shc1 and Dab2. For the macrophage co-IP experiments, membrane proteins from macrophages were prepared as described above after oxLDL treatment (100 µg/ml) for 24 hr.

## shRNA and lentivirus transduction

A pLKO.1 lentiviral vector expressing Shc1 shRNA (TRCN0000055180), with the functional sequence CCGGGCTGAGTATGTTGCCTATGTTCTCGAGAACATAGGCAACATACTCAGCTTTTTG, to target the Shc-1 gene sequence (GCTGAGTATGTTGCCTATGTT), was purchased from Sigma Aldrich (St. Louis, MO). The scramble shRNA was a gift from David Sabatini (Addgene plasmid #1864). (*Sarbassov et al., 2005*) The hairpin sequence was as follows: CCTAAGGTTAAGTCGCCCTCGC TCGAGCGAGGGCGACTTAACCTTAGG.

To generate lentiviral particles, HEK293T cells were co-transfected with the Shc1 shRNA lentiviral plasmid pLKO.1 and Lentiviral packaging mix (psPAX2, pMD2.G; Addgene) using Fugene 6 Transfection Reagent (Roche, Hamburg, German). Culture supernatants containing lentivirus were harvested 48 hr after transfection. For lentiviral transduction, primary macrophage cells were treated with either Shc1-shRNA lentivirus, or scramble shRNA lentivirus for 24 hr. After recovering from viral infection, cells were incubated in fresh RPMI-1640 Medium with 100 µg/mL oxLDL for 24 hr. Cells were then lysed and protein analysis was performed by western blotting.

## Statistics

All values were expressed as mean ±SEM. Unpaired 2-tailed student's t test or two-way ANOVA test were used for statistical analysis. A *P* value less than 0.05 between two groups was considered significant.

## Study approval

Animal procedures were performed according to protocols approved by the Institutional Animal Care and Use Committee (IACUC) at the University of Texas Southwestern Medical Center at Dallas.

## Acknowledgements

This work was supported by NIH grant R37 HL063762, R01 NS093382 and RF1 AG053391 (to JH). JH is further supported by the Consortium for Frontotemporal Dementia Research and the Bright Focus Foundation. ST is supported by North Texas Veterans Affairs Health Care Systems and the American Surgical Association Foundation Award. PB is supported by grants from Fondation de France, Fondation pour la Recherche Médicale (FRM), the Agence Nationale de la Recherche (ANR-06-Physio-032–01 and ANR-09-BLAN-0121–01). JSP is supported by grants from National Institute of Health R01 HL119983 and R01 HL119962. We are indebted to Hoang Dinh Huynh and Yihong Wan for assistance with the bone marrow transplant model; Rebekah Hewitt, Huichuan Reyna, Issac Rocha, Tamara Terrones, Sandy Turner, Grant Richards, John Shelton, Nancy Heard and Barbara Dacus for their excellent technical assistance.

## Additional information

### Funding

| Funder | Grant reference number | Author |
|---|---|---|
| National Institutes of Health | R01 HL119983 | John S Parks |
| National Institutes of Health | R01 HL119962 | John S Parks |
| Fondation de France | | Philippe Boucher |
| Fondation pour la Recherche Médicale | | Philippe Boucher |
| Agence Nationale de la Recherche | ANR-06-Physio-032–01 | Philippe Boucher |
| Agence Nationale de la Recherche | ANR-09-BLAN-0121–01 | Philippe Boucher |
| North Texas Veterans Affairs Health Care Systems | | Shirling Tsai |
| American Surgical Association Foundation | Foundation Award | Shirling Tsai |
| National Institutes of Health | R37 HL063762 | Joachim Herz |
| National Institutes of Health | R01 NS093382 | Joachim Herz |
| National Institutes of Health | RF1 AG053391 | Joachim Herz |
| Consortium for Frontotemporal Dementia Research | | Joachim Herz |
| Bright Focus Foundation | | Joachim Herz |

The funders had no role in study design, data collection and interpretation, or the decision to submit the work for publication.

### Author contributions

Xunde Xian, Conceptualization, Resources, Data curation, Formal analysis, Supervision, Funding acquisition, Validation, Investigation, Visualization, Methodology, Writing—original draft, Project administration, Writing—review and editing; Yinyuan Ding, Conceptualization, Resources, Data curation, Formal analysis, Validation, Investigation, Methodology, Writing—original draft, Project administration, Writing—review and editing; Marco Dieckmann, Conceptualization, Resources, Data curation, Formal analysis, Validation, Investigation, Visualization, Methodology, Writing—original draft, Writing—review and editing; Li Zhou, Conceptualization, Data curation, Formal analysis,

Investigation, Methodology; Florian Plattner, Conceptualization, Resources, Data curation, Investigation, Methodology; Mingxia Liu, Formal analysis, Validation, Writing—review and editing; John S Parks, Resources; Robert E Hammer, Resources, Writing—review and editing; Philippe Boucher, Resources, Methodology; Shirling Tsai, Conceptualization, Formal analysis, Investigation, Methodology, Writing—review and editing; Joachim Herz, Formal analysis, Validation, Writing—original draft, Writing—review and editing

## Author ORCIDs

Xunde Xian (ID) https://orcid.org/0000-0003-3059-1254
Yinyuan Ding (ID) https://orcid.org/0000-0003-4918-8457
Florian Plattner (ID) https://orcid.org/0000-0002-3150-1866
John S Parks (ID) http://orcid.org/0000-0002-5227-8915
Joachim Herz (ID) http://orcid.org/0000-0002-8506-3400

## Ethics

Animal experimentation: Animal procedures were performed according to protocols approved by the Institutional Animal Care and Use Committee (IACUC) at the University of Texas Southwestern Medical Center at Dallas.

## Decision letter and Author response

Decision letter https://doi.org/10.7554/eLife.29292.018
Author response https://doi.org/10.7554/eLife.29292.019

## Additional files

### Supplementary files

• Transparent reporting form
DOI: https://doi.org/10.7554/eLife.29292.017

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
