## [Decision Letter]

[Editors’ note: a previous version of this study was rejected after peer review, but the authors submitted for reconsideration. The first decision letter after peer review is shown below.]

Thank you for choosing to send your work entitled "Tyrosine phosphorylation of the LRP1 intracellular domain regulates macrophage ABCA1 expression" for consideration at *eLife*. Your full submission has been evaluated by Gary Westbrook (Senior editor) and three peer reviewers, one of whom is a member of our Board of Reviewing Editors, and the decision was reached after discussions between the reviewers. Based on our discussions and the individual reviews below, we regret to inform you that your work will not be considered further for publication in *eLife*.

The reviewers appreciated the effort involved in the creation of the Lrp-1 mutant knock-in mice, and commented that the work described was of high quality. However, the consensus was that the results at the present stage are more appropriate for a specialty journal than for the general audience of *eLife*. The reviewers agreed that it was not possible to conclude definitively that macrophages were the cause of the phenotype described and that more direct assessment of macrophage lipid handling was needed. They also believed that more mechanistic insight into the connections between Lrp-1, ppar, lxr and Abca1 would be required for the paper to be appropriate for *eLife*.

Reviewer #1:

In this manuscript, the authors created a knock in mouse model expressing LRP1 lacking tyrosine phosphorylation (Y63F) in its NPxY motif. Mice harboring this mutation exhibited accelerated atherosclerosis. Macrophages isolated from these knock in animals had higher levels of lipid accumulation, and lower levels of ABCA1 expression. The authors showed that the interaction between Shc and LRP1 was dependent on the phosphorylation of Y63, and appeared to be essential for PI3 Kinase/Akt activation and PPARγ/LXRα driven expression of ABCA1. I have the following concerns regarding this manuscript.

1) More evidence is needed to attribute the atherosclerosis phenotype to the macrophages.

a) The authors showed that the PDGF and TGFβ signaling in smooth muscle cells and the morphology of the aorta appeared to be normal in the Y63F knock in animals, and based on these observations excluded the contribution of smooth muscle cells to atherosclerosis. However, it is not known whether or not the Y63F mutation would lead to changes in smooth muscle cell function independent of the PDGF and TGFβ signaling pathways.

b) In addition, the cholesterol metabolism in other tissues such as liver, intestine and adipose tissue was not characterized at all. It is not explained why the Y63F animals have higher plasma cholesterol. It is not clear whether these animals are more prone to developing atherosclerosis just because they have higher plasma LDL? To convincingly claim this is a macrophage caused phenotype, the authors need to do a bone marrow transplant experiment.

2) The authors demonstrated that ABCA1 expression was lower in Y63F macrophages, but they did not provide evidence that ABCA1 is a required factor for the atherosclerosis phenotype. It is not clear whether or not the inflammation and the apoptosis signaling pathways are altered in the Y63F macrophages.

3) There are issues in the characterization of the Y63F macrophages in this study:

a) It is not clear whether or not the expression levels of the Y63F LRP1 are similar to the endogenous LRP1 in macrophages.

b) The interaction between LRP1 and Shc presented in this study is from smooth muscle cells. It is not clear the same is true in macrophages.

c) The uptake and efflux of lipids by the Y63F macrophages need to be assayed, not deduced from the protein levels of several lipid transporters.

4) It is not demonstrated how LRP1 is affecting PPARγ or LXRα transcriptional activity in the present study.

Reviewer #2:

This is a generally interesting set of studies linking LRP1 phosphorylation to macrophage ABCA1 expression/function. The authors propose that the increased atherosclerosis seen in a knock in mouse with a LRP1 that could not be phosphorylated on Y63 was due to decreased macrophage cholesterol efflux through ABCA1, leading to cholesteryl accumulation. The manuscript is well written and argued, but there are still some issues/questions:

1) Lipid metabolism issues: A) The plasma non-HDL lipid levels are higher in the knock in mice- how much of the increased atherosclerosis is simply from that factor? One way of testing this possibility would be to perform a correlation analysis combining data from all the mice and examining the relationship between the plasma values and plaque parameters. B) It is curious that the HDL peak is not apparent on the tracings. LDLr-/- mice have relatively normal HDL-C levels. C) Also related to HDL- ABCCA1 plays a key role in HDL biogenesis and since this is a global knock in, there would be an expected decrease in HDL formation when ABCA1 function/level is impaired. Was there evidence for this?

2) The argument that overall there were no effects on PDGF or TGFβ signaling by tyrosine phosphorylation was based on total aortic homogenate. Though not very likely, it is still possible that there were opposite effects in different cell types in the plaques, resulting in no net changes. Are there any immunohistochemical data on some accepted marker of these pathways to show no changes in SMC and macrophages?

3) Is there a mechanistic basis for the change in necrotic core. Based on the studies of Tabas and others, this may reflect the efferocytotic activity of macrophages- is there evidence in vitro or in vivo for a change in efferocytosis and a mechanistic basis for this?

4) ABCA1 function is also sensitive to its subcellular distribution. In either the in vitro or in vivo macrophages, is there a redistribution of ABCA1 related to LRP1 phosphorylation?

5) Are there other LXR/PPAR targets effected besides ABCA1?- in other words, how specific are the effects? One obvious one is apoE, which is also a macrophage efflux factor.

6) I think a valuable addition would be to use macrophages with the WT and mutant LRP1 in the "Rothblat-Rader" assay in which they are loaded ex vivo with cholesterol (labeled and cold), then injected ip into mice and the integrated RCT pathway assessed by fecal sterol excretion. The host mice should be both WT and knock in, to see if the effects on ABCA1 are at the level of the liver (for HDL formation) or macrophage (for efflux).

Reviewer #3:

The authors generated mice engineered to express LRP1 with impaired tyrosine phosphorylation of the cytoplasmic domain. These mice show increased atherosclerosis after western-type diet with increased plasma cholesterol and TG levels, and increased foam cell accumulation and necrosis in aortic lesions. The authors show that lack of tyrosine phosphorylation in macrophage LRP1 reduces ABCA1 expression by interrupting Shc-1 binding and AKT phosphorylation in vitro. They conclude that tyrosine phosphorylation of LRP1 in macrophages regulates cholesterol efflux and is essential to prevent atherosclerosis.

1) Increased atherosclerosis may simply be the consequence of differences in cholesterol and TG. Although the authors showed connections between impaired macrophage LRP1 tyrosine phosphorylation and ABCA1 expression, has reduced ABCA1 expression been shown in vivo in the mouse to cause atherosclerosis independent of plasma lipid levels?

2) The authors claim that LRP1 regulates cholesterol efflux. However, experiments to test efflux capacity by macrophages defective in LRP1 phosphorylation were not performed.

3) The claim that the lack of phosphorylation of LRP1 in SMC does not affect SMC function is not corroborated by data on function, as all they show is IP-WB and in vivo staining of SMC in the arterial wall.

4) Data of Figure 9 are less convincing than reported, and do not provide a mechanistic proof.

[Editors’ note: what now follows is the decision letter after the authors submitted for further consideration.]

Thank you for submitting your work entitled "Phosphorylation of the LRP1 NPxY motif protects against atherosclerosis by regulating macrophage cholesterol export" for consideration by *eLife*. Your article has been reviewed by three peer reviewers, and the evaluation has been overseen by a Reviewing Editor and a Senior Editor. The reviewers have opted to remain anonymous. Our decision has been reached after consultation between the reviewers. Based on these discussions and the individual reviews below, we regret to inform you that your work will not be considered further for publication in *eLife*.

Summary

The reviewers recognized and applauded the considerable amount of experimental data in this paper, but they were skeptical that the significant decrease in atherosclerosis in the mutant LRP mouse could be explained by changes in ABCA1 expression. Studies by the laboratory of John Parks have suggested that the impact of macrophage ABCA1 deficiency on atherosclerosis are modest. Addressing this issue in a definitive fashion would necessitate breeding double knockout mice. Also, the reviewers thought that it would be important to test whether restoring AKT expression would restore the perturbed expression of ABCA1. In online discussions, the reviewers suggested that the impact of the LRP mutation on efferocytosis was not well developed. Finally, multiple reviewers thought that the paper might be better suited for a specialty journal.

Reviewer #2:

The authors describe an elegant approach to analyzing the functions of the NPxY motif in the intracellular domain of LRP1 and uncover a number of interesting findings, including well defined lack of Shc1 binding, alterations in plasma lipoproteins and increases in atherosclerosis when mice are challenged with high fat/cholesterol/bile salt diet. Moreover, the increase in atherosclerosis is also seen in a bone marrow transplantation study. The authors also show that the mutation reduces Abca1 expression and impairs AKT responses to oxidized LDL in peritoneal macrophages. However, the main conclusion as stated in the title of the Abstract is not well supported by the data provided.

1) There is about a 40% reduction in Abca1 mRNA an protein in LRP mutant macrophages. What is Abca1 expression in plaques? The authors cite Aiello et al. who did an ABCA1 bone marrow transplantation study as evidence that macrophage ABCA1 deficiency is sufficient to explain the atherosclerosis phenotype. However, studies which more specifically examined the impact of myeloid/macrophage deficiency of ABCA1 (as opposed to whole bone marrow deficiency) revealed at best a minimal atherosclerosis phenotype (Bi et al., 2014). Thus it is unlikely that reduced macrophage Abca1 can explain the fairly dramatic atherosclerosis findings in the present study. The authors may wish to consider making a compound mouse with both the LRP1 mutation and ABCA1 deficiency if they wish to examine this issue more critically.

2) While it seems clear that the LRP1 mutation causes a reduced AKT response to oxLDL it is less clear that this causes the reduction in Abca1. The authors use high concentrations of AKT inhibitors to show reductions in Abca1 and there is concern about cell toxicity. In any event, a more informative experiment would be to restore AKT in LRP1 mutant cells (e.g. using Myr-AKT) to see if that reverses the defect in Abca1.

3) Information on sex, genetic background and housing of mice (littermates together?) does not seem to be provided. This is especially important for atherosclerosis studies.

4) Necrotic core area increase is proportional to the overall increase in lesion size. More direct methods to assess efferocytosis in plaques could be used if the authors feel this is an important part of the paper.

Reviewer #3:

I read carefully the responses to my original comments and also read the responses to the other reviewers. In general, the authors have tried to be very responsive, especially in regard to providing data to support their points and to address the issues raised. I feel that in the area of efferocytosis, some rigor is still lacking. They note that MerTK expression is lower, and state that the decrease in efferocytosis is likely attributed to Mertk dysfunction. Likely is not a direct link, and even if this were true, the mechanism linking the mutation in Lrp1 to decreased MerTK expression and possible function is not shown.

Reviewer #4:

The authors have addressed all of the comments I had raised and have produced a monumentally large revised manuscript that challenges the reader's attention span and is overly rich in figures. Figure 4, Figure 6 and panel B of Figure 13 are not essential.

---

## [Author Response]

[Editors’ note: the author responses to the first round of peer review follow.]

Reviewer #1:In this manuscript, the authors created a knock in mouse model expressing LRP1 lacking tyrosine phosphorylation (Y63F) in its NPxY motif. Mice harboring this mutation exhibited accelerated atherosclerosis. Macrophages isolated from these knock in animals had higher levels of lipid accumulation, and lower levels of ABCA1 expression. The authors showed that the interaction between Shc and LRP1 was dependent on the phosphorylation of Y63, and appeared to be essential for PI3 Kinase/Akt activation and PPARγ/LXRα driven expression of ABCA1. I have the following concerns regarding this manuscript.1) More evidence is needed to attribute the atherosclerosis phenotype to the macrophages.a) The authors showed that the PDGF and TGFβ signaling in smooth muscle cells and the morphology of the aorta appeared to be normal in the Y63F knock in animals, and based on these observations excluded the contribution of smooth muscle cells to atherosclerosis. However, it is not known whether or not the Y63F mutation would lead to changes in smooth muscle cell function independent of the PDGF and TGFβ signaling pathways.

Atherosclerosis is associated with a phenotypic switch in SMCs from a contractile to a synthetic phenotype, characterized by increased SMC migration and proliferation regulated by growth promoters such as PGFβ, TGFβ, endothelin-1 (ET-1), thrombin, FGF, IL-1 and nitric oxide (NO). Previous studies in SMCs demonstrated that LRP1 was critical for PDGF-induced cell proliferation and migration. We initially hypothesized that this may be dependent upon tyrosine phosphorylation of the LRP1 cytoplasmic domain. Therefore, we used the Transwell Migration assay and BrdU Proliferation Assay to compare the migratory and proliferative response of primary aortic smooth muscle cells explanted from wild type or LRP1^Y63F^ mice to PDGFβ or TGFβ. As shown in Author response image 1, no difference in migration (A) or proliferation (B) was observed betweenwild type and LRP1^Y63F^ smooth muscle cells.

This figure has been included in the manuscript as Figure 4.

b) In addition, the cholesterol metabolism in other tissues such as liver, intestine and adipose tissue was not characterized at all. It is not explained why the Y63F animals have higher plasma cholesterol. It is not clear whether these animals are more prone to developing atherosclerosis just because they have higher plasma LDL? To convincingly claim this is a macrophage caused phenotype, the authors need to do a bone marrow transplant experiment.

As shown in Figure 2, there was a slight increase in plasma cholesterol levels in LRP1^Y63F^;LDLR^-/-^ on HFHC diet. To understand the source of elevated total cholesterol, we performed FPLC lipoprotein analysis, which showed that elevated plasma cholesterol levels in LRP1^Y63F^;LDLR^-/-^ mice fed with HCHF diet were attributed to increased VLDL-cholesterol with slightly reduced LDL- and HDL-cholesterol contents (Author response image 1). Moreover, plasma apoB48 concentrations were markedly increased, with a shift toward larger sizes of VLDL/LDL-containing fractions in LRP1^Y63F^;LDLR^-/-^ mice compared to LDLR^-/-^ mice on HCHF diet. These results suggested that the LRP1 Y63F mutation resulted in a modest decreased clearance of postprandial lipids.

To definitively demonstrate that the proatherogenic phenotype is due to macrophage pathology rather than elevated plasma cholesterol levels, we generated a bone marrow transplant model wherein LDLR^-/-^ or LRP1^Y63F^;LDLR^-/-^ bone-marrow derived mononuclear cells were transplanted into irradiated LDLR^-/-^ mice. These mice were then fed a HCHF diet for 16 weeks.

As shown in Figure 6, HCHF diet resulted in very high plasma cholesterol levels after radiation in all mice. However, there was no difference in plasma lipid and apolipoprotien levels between LDLR^-/^→LDLR^-/-^ and LRP1^Y63F^;LDLR^-/-^→LDLR^-/-^ mice (Author response image 1, Figure 6). By comparing Figure 6 to Figure 2 (Author response image 1), we demonstrated that bone marrow transplantation eliminated the difference in FPLC lipid profiles. This suggested that the difference in lipid profiles seen in LDLR^-/-^ vs. LRP1^Y63F^;LDLR^-/-^ mice is due to the effect of the systemic LRP1^Y63F^ mutation in the liver. Since LRP1 is involved in hepatic chylomicron clearance as well as ABCA1 expression, and ABCA1 is substantially reduced in Y63F mutant livers (Figure 7), we suspect that the modest difference in the lipoprotein profiles is mostly due to these mutation-induced changes of hepatic LPR1 function and the concomitant reduction in ABCA1 expression in the liver, which generates a hypomorphic Tangier phenotype. The molecular mechanism leading to these changes in liver function will require further extensive studies, which will become the focus of one or more independent manuscripts.

As shown in Figure 8, despite the dramatically elevated cholesterol levels in both LDLR^-/-^→LDLR^-/-^ and LRP1^Y63F^;LDLR^-/-^→LDLR^-/-^ mice on HCHF diet, only the LRP1^Y63F^;LDLR^-/-^→LDLR^-/-^ on HCHF diet developed severe atherosclerosis (Author response image 2). These results demonstrate that the LRP1^Y63F^ mutation in macrophages is sufficient to cause the observed exaggerated atherosclerotic response, independent of the modestly elevated plasma cholesterol levels in the systemic mutant.

2) The authors demonstrated that ABCA1 expression was lower in Y63F macrophages, but they did not provide evidence that ABCA1 is a required factor for the atherosclerosis phenotype. It is not clear whether or not the inflammation and the apoptosis signaling pathways are altered in the Y63F macrophages.

Recent studies with genetically engineered mice lacking or overexpressing ABCA1 provide evidence that ABCA1 modulates atherosclerosis susceptibility. For example, ApoE^-/-^ mice lacking ABCA1 specifically in liver (HSKO), developed significantly larger atherosclerotic lesions than control apoE^-/-^ mice after 12 weeks on chow diet, suggesting that hepatic ABCA1 is antiatherogenic.

1). Brunham LR1, Singaraja RR, Duong M, Timmins JM, Fievet C, Bissada N, Kang MH, Samra A, Fruchart JC, McManus B, Staels B, Parks JS, Hayden MR. Tissue-specific roles of ABCA1 influence susceptibility to atherosclerosis.Arterioscler Thromb Vasc Biol. 2009 Apr;29(4):548-54.

Furthermore, specific disruption of ABCA1 in bone marrow–derived cells induced atherosclerotic lesion development whereas macrophage overexpression of ABCA1 resulted in decreased susceptibility to spontaneous atherosclerosis in apoE knockout mice and in C57BL/6 mice with diet-induced atherosclerosis.

2). Van Eck M1, Bos IS, Kaminski WE, Orsó E, Rothe G, Twisk J, Böttcher A, Van Amersfoort ES, Christiansen-Weber TA, Fung-Leung WP, Van Berkel TJ, Schmitz. Leukocyte ABCA1 controls susceptibility to atherosclerosis and macrophage recruitment into tissues. Proc Natl Acad Sci U S A. 2002 Apr 30;99(9):6298-303. Epub 2002 Apr 23.

3). Van Eck M1, Singaraja RR, Ye D, Hildebrand RB, James ER, Hayden MR, Van Berkel TJ. Macrophage ATP-binding cassette transporter A1 overexpression inhibits atherosclerotic lesion progression in low-density lipoprotein receptor knockout mice. Arterioscler Thromb Vasc Biol. 2006 Apr;26(4):929-34.

As shown in Figure 5, TUNEL staining of aortic root sections from HCHF-fed LDLR^-/-^ and LRP1^Y63F^;LDLR^/-^ mice or oxLDL-treated peritoneal macrophages from LDLR^-/-^ and LRP1^Y63F^;LDLR^-/-^ mice show that LRP1^Y63F^;LDLR^-/-^ mice have increased numbers of apoptotic cells. (Author response image 2). Previous studies have demonstrated that deficiency in Mertk is associated with increased macrophage apoptosis and decreased efferocytosis, leading to increased necrotic core in Apoe^-/-^ mice. Similarly, we have shown that peritoneal macrophages from LRP1^Y63F^;LDLR^-/-^ macrophages have decreased expression of Mertk both at baseline and in response to treatment with oxLDL (Author response image 2). Therefore, increased necrotic core and apoptosis in the LRP1^Y63F^;LDLR^-/-^ mice is likely attributed to Mertk dysfunction caused by the macrophage LRP1 mutation.

4). Scott RS1, McMahon EJ, Pop SM, Reap EA, Caricchio R, Cohen PL, Earp HS, Matsushima GK. Phagocytosis and clearance of apoptotic cells is mediated by MER. Nature. 2001 May 10;411(6834):207-11.

5). Li Y, Gerbod-Giannone MC, Seitz H, Cui D, Thorp E, Tall AR, Matsushima GK, Tabas I. Cholesterolinduced apoptotic macrophages elicit an inflammatory response in phagocytes, which is partially attenuated by the Mer receptor. J Biol Chem. 2006 Mar 10;281(10):6707-17. Epub 2005 Dec 27.

6). Thorp E1, Cui D, Schrijvers DM, Kuriakose G, Tabas I. Mertk receptor mutation reduces efferocytosis efficiency and promotes apoptotic cell accumulation and plaque necrosis in atherosclerotic lesions of apoe^-/-^ mice. Arterioscler Thromb Vasc Biol. 2008 Aug;28(8):1421-8.

**Author response image 2. respfig2:** 

3) There are issues in the characterization of the Y63F macrophages in this study:a) It is not clear whether or not the expression levels of the Y63F LRP1 are similar to the endogenous LRP1 in macrophages.

The endogenous LRP1 expression levels in LRP1^Y63F^;LDLR^-/-^ macrophages were identical to those in the LDLR^-/-^ macrophages. This data is also summarized in Figure 9.

b) The interaction between LRP1 and Shc presented in this study is from smooth muscle cells. It is not clear the same is true in macrophages.

The co-immunoprecipitation was repeated in isolated peritoneal macrophages. Peritoneal macrophages were treated with oxLDL (100µg/mL) for 24 hours, followed by lysis and coimmunoprecipation. Similar to the case in SMCs, we found that the Y63F mutation impaired association of Shc1 with LRP1. This data has been included in Figure 13.

c) The uptake and efflux of lipids by the Y63F macrophages need to be assayed, not deduced from the protein levels of several lipid transporters.

To assess cholesterol efflux in macrophages, we performed cholesterol efflux experiments in peritoneal macrophages from LDLR^-/-^ and LRP1^Y63F^;LDLR^-/-^ mice using [^[3]^H] cholesterol. Compared with LDLR^-/-^ macrophages, LRP1^Y63F^;LDLR^-/-^ macrophages showed significantly reduced efflux [E%] of [^[3]^H] cholesterol to apoAI but not to HDL. This may contribute to increased intracellular lipid accumulation in LRP1^Y63F^;LDLR^-/-^ macrophages when incubated with oxLDL. This data has been included in Figure 9.

4) It is not demonstrated how LRP1 is affecting PPARγ or LXRα transcriptional activity in the present study.

Our data in Figure 9–Figure 13 show that the LRP1^Y63F^ mutation affects LXR and PPARγ activity, and that this is dependent on the upstream phosphorylation of Akt. Previous work by Andre Tremblay’s group revealed that activation of the PI3K/Akt pathway leads to Akt recruitment to PPARγ, leading to increased transcriptional activity of PPARγ and LXRα and subsequent increased expression of ABCA1.

1) Demers A1, Caron V, Rodrigue-Way A, Wahli W, Ong H, Tremblay A. A concerted kinase interplay identifies PPARgamma as a molecular target of ghrelin signaling in macrophages. PLoS One. 2009 Nov 4;4(11):e7728.

Our studies first showed that LRP1^Y63F^;LDLR^-/-^ mice had lower basal levels of ABCA1 and phosphorylated Akt levels both in the presence and absence of oxLDL (Figure 9 and Figure 12). We then determined that ABCA1 expression was increased by Rosiglitazone, T0901317, and LXR623 in a dose dependent manner in both LDLR^-/-^ and LRP1^Y63F^;LDLR^-/-^ macrophages. However, rescue with LXR and PPARγ agonists did not fully correct the blunted ABCA1 levels in the LRP1^Y63F^;LDLR^-/-^ mice (Figure 11). We further demonstrated that inhibition of PI3K/Akt results in incomplete induction of ABCA1 by both PPARγ (Figure 11) and LXR agonists (Figure 11), similar to what is observed in macrophages with the LRP1Y63F mutation (Figure 11). Finally, we confirmed that binding of Shc1 to LRP1 was required for Akt phosphorylation in macrophages (Figure 13). The LRP1^Y63F^ mutation, which precludes phosphorylation of the second NPxY motif of LRP1 and thus inhibits binding of Shc1, leads to reduced Akt phosphorylation in response to oxLDL. Taken together, the decreased induction of ABCA1 in response to oxLDL in LRP1^Y63F^;LDLR^-/-^ macrophages is due to reduced Akt phosphorylation, which in turn modulates PPARγ/LXR-driven gene expression.

Reviewer #2:This is a generally interesting set of studies linking LRP1 phosphorylation to macrophage ABCA1 expression/function. The authors propose that the increased atherosclerosis seen in a knock in mouse with a LRP1 that could not be phosphorylated on Y63 was due to decreased macrophage cholesterol efflux through ABCA1, leading to cholesteryl accumulation. The manuscript is well written and argued, but there are still some issues/questions:1) Lipid metabolism issues: A) The plasma non-HDL lipid levels are higher in the knock in mice- how much of the increased atherosclerosis is simply from that factor? One way of testing this possibility would be to perform a correlation analysis combining data from all the mice and examining the relationship between the plasma values and plaque parameters. B) It is curious that the HDL peak is not apparent on the tracings. LDLr-/- mice have relatively normal HDL-C levels. C) Also related to HDL- ABCCA1 plays a key role in HDL biogenesis and since this is a global knock in, there would be an expected decrease in HDL formation when ABCA1 function/level is impaired. Was there evidence for this?

A) See reviewer #1 comments 1b.

B) We collected fresh plasma from additional mice and repeated the FPLC experiments. As shown in the Author response image 1, LDLR^-/-^ mice have a higher HDL-C peak than LRP1^Y63F^;LDLR^-/-^ mice (A) while there is no difference of HDL-C levels between two bone marrow transplantation groups (B). This suggests that the small differences in lipid profile between LDLR^-/-^ and LRP1^Y63F^;LDLR^-/-^ mice are due to effects of the systemic LRP1^Y63F^ mutation. Since LRP1 is essential in hepatic chylomicron clearance and also for hepatic ABCA1 expression (Figure 7), we conclude that the Y63F mutation may affect hepatic clearance of VLDL as well as HDL catabolism, thus accounting for the difference between LDLR^-/-^ and LRP1^Y63F^;LDLR^-/-^ mice and the bone marrow transplant mice. These corrected data are included in Figure 2.

C) To support our hypothesis that the changes in lipoprotein profile are due to deficiency in hepatic LRP1, we also found that hepatic ABCA1 levels were significantly reduced in LRP1^Y63F^;LDLR^-/-^ mice compared to LDLR^-/-^ mice (Figure 7, panel A), while there was no difference between two BMT groups (Figure 7, panel B).

2) The argument that overall there were no effects on PDGF or TGFβ signaling by tyrosine phosphorylation was based on total aortic homogenate. Though not very likely, it is still possible that there were opposite effects in different cell types in the plaques, resulting in no net changes. Are there any immunohistochemical data on some accepted marker of these pathways to show no changes in SMC and macrophages?

We appreciatethe comment that there may be cell-type dependent differences. This has also now been included into the discussion. In Figure 4, we found no difference in markers of PDGF signaling in total aortic homogenate. In Figure 4, we demonstrated that there was no difference in PDGF mediated signaling in smooth muscle cells explanted from the respective mouse aortas. We have focused primarily on PDGF mediated signaling, since this pathway, leading to mitogenic signaling, has been implicated in atherosclerosis.

3) Is there a mechanistic basis for the change in necrotic core. Based on the studies of Tabas and others, this may reflect the efferocytotic activity of macrophages- is there evidence in vitro or in vivo for a change in efferocytosis and a mechanistic basis for this?

See reviewer #1 comment 2.

4) ABCA1 function is also sensitive to its subcellular distribution. In either the in vitro or in vivo macrophages, is there a redistribution of ABCA1 related to LRP1 phosphorylation?

The ratio of surface/total ABCA1 in LRP1^Y63F^;LDLR^-/-^ macrophages was similar to that in LDLR^-/-^ microphages, although the levels of surface and total ABCA1 were both reduced in LRP1^Y63F^;LDLR^-/-^ macrophages. This suggests that the LRP1^Y63F^ mutation results in decreased ABCA1 expression rather than deficiency of ABCA1 transport to the cell surface. This data has been included in Figure 9.

5) Are there other LXR/PPAR targets effected besides ABCA1?- in other words, how specific are the effects? One obvious one is apoE, which is also a macrophage efflux factor.

As shown in the Author response image 3,after incubation with oxLDL for 24 hours, the mRNA levels of Abca1 and Lpl were significantly reduced in LRP1^Y63F^;LDLR^-/-^ macrophages, with a similar trend for a reduction of Apoe mRNA levels that, however, did not reach significance, compared to LDLR^-/-^ microphages. All 3 genes are downstream targets of LXR. However, Abca1 expression and ABCA1mediated cholesterol export showed the most dramatic changes in response to LRP1 mutation and oxLDL, and this effect is confirmed at the protein expression level by western blot (Figure 9).

**Author response image 3. respfig3:** 

6) I think a valuable addition would be to use macrophages with the WT and mutant LRP1 in the "Rothblat-Rader" assay in which they are loaded ex vivo with cholesterol (labeled and cold), then injected ip into mice and the integrated RCT pathway assessed by fecal sterol excretion. The host mice should be both WT and knock in, to see if the effects on ABCA1 are at the level of the liver (for HDL formation) or macrophage (for efflux).

These points have been addressed by the cholesterol efflux study (please see reviewer #1 comment 3c) and the bone marrow transplant model (please see reviewer #1 comment 1b).

Reviewer #3:The authors generated mice engineered to express LRP1 with impaired tyrosine phosphorylation of the cytoplasmic domain. These mice show increased atherosclerosis after western-type diet with increased plasma cholesterol and TG levels, and increased foam cell accumulation and necrosis in aortic lesions. The authors show that lack of tyrosine phosphorylation in macrophage LRP1 reduces ABCA1 expression by interrupting Shc-1 binding and AKT phosphorylation in vitro. They conclude that tyrosine phosphorylation of LRP1 in macrophages regulates cholesterol efflux and is essential to prevent atherosclerosis.1) Increased atherosclerosis may simply be the consequence of differences in cholesterol and TG. Although the authors showed connections between impaired macrophage LRP1 tyrosine phosphorylation and ABCA1 expression, has reduced ABCA1 expression been shown in vivo in the mouse to cause atherosclerosis independent of plasma lipid levels?

See responses above and reviewer #1 comments 1b.

2) The authors claim that LRP1 regulates cholesterol efflux. However, experiments to test efflux capacity by macrophages defective in LRP1 phosphorylation were not performed.

See responses above and reviewer #1 comment 3c.

3) The claim that the lack of phosphorylation of LRP1 in SMC does not affect SMC function is not corroborated by data on function, as all they show is IP-WB and in vivo staining of SMC in the arterial wall.

See responses above and reviewer #1 comment 1a.

4) Data of Figure 9 are less convincing than reported, and do not provide a mechanistic proof.

We have performed additional experiments to demonstrate that the Y63F mutation inhibits binding of Shc1 to LRP1 in macrophages (Figure 13) and have proposed a mechanism consistent with our data and the relevant literature in Figure 14. LRP1 tyrosine phosphorylation is necessary for interaction with Shc1, which in turn activates PI3K/Akt signaling. Phospho-Akt modulates PPARγ/LXR driven Abca1 gene transcription and is necessary for the full effect of ox-LDL and PPARγ/LXR induced expression of ABCA1 to mediate cholesterol efflux. Inhibition of PI3K/Akt (using LY294002 or Akt inhibitor) results in only partial induction of Abca1 in response to ox-LDL and PPARγ/LXR agonists (rosiglitazone, T0901317, LXR623). (B) In the absence of LRP1 tyrosine phosphorylation, as in the case of the LRP1Y63F mutation, Dab2 rather than Shc1 associates with LRP1. Lack of the interaction with Shc1 decreases p-Akt, which in turn decreases Abca1 gene transcription, even in the presence of ox-LDL and PPARγ/LXR agonists, similar to what is observed in LRP1 wild type cells treated with PI3K/Akt inhibitors. Ultimately, decreased ABCA1 expression leads to impaired cholesterol efflux, thus accelerating atherogenesis.

[Editors' note: the author responses to the re-review follow.]

SummaryThe reviewers recognized and applauded the considerable amount of experimental data in this paper, but they were skeptical that the significant decrease in atherosclerosis in the mutant LRP mouse could be explained by changes in ABCA1 expression. Studies by the laboratory of John Parks have suggested that the impact of macrophage ABCA1 deficiency on atherosclerosis are modest. Addressing this issue in a definitive fashion would necessitate breeding double knockout mice. Also, the reviewers thought that it would be important to test whether restoring AKT expression would restore the perturbed expression of ABCA1. In online discussions, the reviewers suggested that the impact of the LRP mutation on efferocytosis was not well developed. Finally, multiple reviewers thought that the paper might be better suited for a specialty journal.Reviewer #2:The authors describe an elegant approach to analyzing the functions of the NPxY motif in the intracellular domain of LRP1 and uncover a number of interesting findings, including well defined lack of Shc1 binding, alterations in plasma lipoproteins and increases in atherosclerosis when mice are challenged with high fat/cholesterol/bile salt diet. Moreover, the increase in atherosclerosis is also seen in a bone marrow transplantation study. The authors also show that the mutation reduces Abca1 expression and impairs AKT responses to oxidized LDL in peritoneal macrophages. However, the main conclusion as stated in the title of the Abstract is not well supported by the data provided.1) There is about a 40% reduction in Abca1 mRNA an protein in LRP mutant macrophages. What is Abca1 expression in plaques? The authors cite Aiello et al. who did an ABCA1 bone marrow transplantation study as evidence that macrophage ABCA1 deficiency is sufficient to explain the atherosclerosis phenotype. However, studies which more specifically examined the impact of myeloid/macrophage deficiency of ABCA1 (as opposed to whole bone marrow deficiency) revealed at best a minimal atherosclerosis phenotype (Bi et al., 2014). Thus it is unlikely that reduced macrophage Abca1 can explain the fairly dramatic atherosclerosis findings in the present study. The authors may wish to consider making a compound mouse with both the LRP1 mutation and ABCA1 deficiency if they wish to examine this issue more critically.2) While it seems clear that the LRP1 mutation causes a reduced AKT response to oxLDL it is less clear that this causes the reduction in Abca1. The authors use high concentrations of AKT inhibitors to show reductions in Abca1 and there is concern about cell toxicity. In any event, a more informative experiment would be to restore AKT in LRP1 mutant cells (e.g. using Myr-AKT) to see if that reverses the defect in Abca1.3) Information on sex, genetic background and housing of mice (littermates together?) does not seem to be provided. This is especially important for atherosclerosis studies.4) Necrotic core area increase is proportional to the overall increase in lesion size. More direct methods to assess efferocytosis in plaques could be used if the authors feel this is an important part of the paper.Reviewer #3:I read carefully the responses to my original comments and also read the responses to the other reviewers. In general, the authors have tried to be very responsive, especially in regard to providing data to support their points and to address the issues raised. I feel that in the area of efferocytosis, some rigor is still lacking. They note that MerTK expression is lower, and state that the decrease in efferocytosis is likely attributed to Mertk dysfunction. Likely is not a direct link, and even if this were true, the mechanism linking the mutation in Lrp1 to decreased MerTK expression and possible function is not shown.Reviewer #4:The authors have addressed all of the comments I had raised and have produced a monumentally large revised manuscript that challenges the reader's attention span and is overly rich in figures. Figure 4, Figure 6 and panel B of Figure 13 are not essential.

First, I want to thank you for giving us the opportunity to resubmit our manuscript formerly entitled “Phosphorylation of the LRP1 NPxY motif protects against atherosclerosis by regulating macrophage cholesterol export” and now entitled “LRP1 integrates macrophage cholesterol homeostasis and inflammatory responses in atherosclerosis”. Next, I should apologize if you found the tone of my previous letter not appropriate or helpful. It was borne out of the frustrations I and my coauthors felt as we were trying to make sense of the erratic appearance of the reviewers’ criticism, who appeared to question the role of ABCA1 in reverse cholesterol transport and atherosclerosis while every single published paper we were able to find reported the exact opposite, consistent with our own findings. Your editorial summary has now made this a lot clearer and we have revised the manuscript along these lines, after reconfirming and adding additional data to address specific reviewers’ concerns and questions. In addition, and specifically in order to address the misconception that macrophage ABCA1 plays no role in atherosclerosis, I am attaching a clarifying letter by my colleague and coauthor, John Parks, who also is the senior author of the Bi et al. study on which the reviewers had based their assessment. We hope that letter clarifies that our conclusions are in no way in conflict with the Bi et al. study. At this point let me emphasize again: The main thrust of our present manuscript was not to investigate the role of ABCA1 in atherosclerosis, which we had considered a settled issue and for which we have abundantly cited the pertinent literature. The main point of our study was to investigate the consequences of the tyrosine phosphorylation in the cytoplasmic domain of LRP1, which we find protects from atherosclerosis by activating a signaling axis that includes Shc1, PI3K, Akt, and PPARγ and which results in the boosting of the atheroprotective expression of LXR target genes. ABCA1 is a major LXR target and one that has been clearly shown to be atheroprotective by itself in numerous independent studies in the mouse and in humans (including in the Bi et al. study). The major difference between the ABCA1 knockout models, which show reduced VLDL production and thus have a less atherogenic plasma lipoprotein profile, and our LRP1 knockin mouse is that the former are lacking ABCA1 completely in their monocytes/macrophages, while in our model ABCA1 expression is reduced with lower, but clearly present, residual cholesterol export activity. Therefore, circulating monocytes, which do not have to deal with the excessive cholesterol-loading that resident macrophages in atherosclerotic lesions have to cope with, would be less prone to convert to an inflammatory phenotype than monocytes that completely lack ABCA1. In the course of being fed an atherogenic diet high in cholesterol the latter – but not the LRP1 mutant monocyte/macrophages that express ABCA1 merely at a lower level – would be unable to export the excess cholesterol taken up from the circulation through passive or active receptor-mediated mechanisms. This resulted in the increased expression of inflammatory cytokines, which reduced VLDL secretion by the liver through mechanisms that remain incompletely understood. In support of this interpretation of the data, Bi and colleagues found that in the absence of excess cholesterol feeding, inflammatory cytokine expression by genetically ABCA1-deficient monocyte/macrophages subsided, resulting in a normal lipoprotein profile, which now unmasked the previously occluded atheroprotective effect of macrophage ABCA1. The Bi et al. paper clearly states this, as the accompanying letter by Dr. Parks also confirms.

We hope you and the reviewers will accept this detailed explanation of what the reviewers felt was an apparent conflict with the prior literature. However, there is actually no conflict. While we could put this detailed explanation into the discussion of the manuscript, we feel that this would merely distract from the main topic of the article, which is the tyrosine phosphorylationdependent LRP1 signaling axis, and send the discussion off onto an irrelevant tangent. We have thus addressed this minor issue only briefly by stating “Our results, which show an identical atherogenic lipoprotein profile upon cholesterol-feeding of LDLR^-/-^ mice transplanted with LDLR^-/-^ or LDLR^-/-^;LRP1^Y63F^ bone marrow, differ from those of Bi et al., who showed reduced plasma lipid levels in mice lacking ABCA1 completely in their monocytes/macrophages. In their model, elevated plasma cholesterol levels and the resulting cholesterol overload led to the secretion of inflammatory cytokines, which subsided on a low cholesterol chow diet. In our case, monocyte/macrophages carrying the Y63F mutation in LRP1 retained residual, though substantially lower, ABCA1 expression, which would protect the circulating cells from cholesterol overload in a hyperlipidemic, cholesterol-fed state.”

In addition to these specific changes, we have substantially rewritten the entire manuscript to emphasize the role of the novel LRP1/Shc/PI3K/Akt/PPARγ/LXR axis in the regulation of atheroprotective gene expression, in complete agreement with the seminal work by Peter Tontonoz and colleagues. We have striven to deemphasize the spotlight which was previous focused perhaps too prominently upon ABCA1 and instead have greatly broadened our discussion and interpretation of our results. In addition, we have discussed the documented roles of LRP1 in the suppression of the inflammatory response and in efferocytosis, as so elegantly described by Ira Tabas and his group. In support of our conclusions, especially the role of ABCA1 for the protection from atherosclerosis, we have also cited the comprehensive work and reviews from Alan Tall and colleagues and from Michael Hayden’s group. A fascinating topic, and one we would love to continue to explore, ideally in collaboration with the Tabas laboratory, is the role of the LRP1 tyrosine phosphorylation in efferocytosis. Given the sheer amount of work we have reported here, however, this is unfortunately far beyond the scope of the present study. However, our new animal model, reported here, will now open the door to explore this avenue.